# Assessment of the theoretical limit in instrumental detectability of northern high-latitude methane sources using $\delta^{13}CH_4$ atmospheric signal

Thibaud Thonat[1], Marielle Saunois[1], Isabelle Pison[1], Antoine Berchet[1], Thomas Hocking[1], Brett F. Thornton[2], Patrick M. Crill[2] and Philippe Bousquet[1]

[1] Laboratoire des Sciences du Climat et de l'Environnement, LSCE/IPSL, CEA-CNRS-UVSQ, Université Paris-Saclay, F-91191 Gif-sur-Yvette, France
[2] Department of Geological Sciences and Bolin Centre for Climate Research, Svante Arrhenius väg 8, 106 91, Stockholm, Sweden

*Correspondence to*: Marielle Saunois (marielle.saunois@lsce.ipsl.fr)

**Abstract.**

Recent efforts brought together bottom-up quantification approaches (inventories, process-based models) and top-down approaches using regional observations of methane atmospheric concentrations through inverse modelling to better estimate the northern high-latitude methane sources. Nevertheless, for both approaches, the relatively small number of available observations in northern high-latitude regions leaves gaps in our understanding of the drivers and distributions of the different types of regional methane sources . Observations of methane isotope ratios, performed with instruments that are becoming increasingly affordable and accurate, could bring new insights on the contributions of methane sources and sinks. Here, we present the source signal that could be observed from methane isotopic $^{13}CH_4$ measurements if high-resolution observations were available, and thus what requirements should be fulfilled in future instrument deployments in terms of accuracy in order to constrain different emission categories. This theoretical study uses the regional chemistry-transport model CHIMERE driven by different scenarios of isotopic signatures for each regional methane source mix. It is found that if the current network of methane monitoring sites were equipped with instruments measuring the isotopic signal continuously, only sites that are significantly influenced by emission sources could differentiate regional emissions with a reasonable level of confidence. For example, wetland emissions require daily accuracies lower than 0.2‰ for most of the sites. Detecting ESAS emissions requires accuracies lower than 0.05‰ at coastal Russian sites (even lower for other sites). Freshwater emissions would be detectable with uncertainty lower than 0.1‰ for most continental sites. Except Yakuskt, Siberian sites require stringent uncertainty (lower than 0.05‰) to detect anthropogenic emissions from oil and gas, or coal production. Remote sites such as Zeppelin, Summit or Alert, requires daily uncertainty below 0.05‰ to detect any regional sources. These limits vary with the hypothesis on isotopic signatures assigned to the different sources.

## 1 Introduction

Atmospheric methane ($CH_4$) is a potent climate forcing gas, responsible for more than 20% of the direct additional radiative forcing caused by human activities since pre-industrial times (Ciais et al., 2013; Etminan et al., 2016). After staying nearly constant between 1999 and 2006, methane concentrations have been increasing again (Dlugokencky et al., 2011; Saunois et al., 2016). The explanations of this renewed accumulation are still widely debated. Recent studies, however, stress the major role played by microbial sources, particularly in the tropics (Schaeffer et al., 2016; Nisbet et al., 2016; McNorton et al., 2016; Saunois et al., 2017) together with uncertain contributions of fossil-fuel-related emissions (Schwietzke et al., 2017; Saunois et al., 2016) associated with a probable decrease in biomass burning emissions (Worden et al., 2018). Decreases in atmospheric sinks (Naus et al., 2019; Rigby et al., 2017; Turner et al., 2016) have also been postulated to contribute to the rise, though changes in methane sink cannot explain this rise by themselves.

Although the northern high-latitudes (>60°N) represent only about 4% of global methane emissions (Saunois et al., 2016) and does not seem to be a main contributor to the increasing trend of the past decade (e.g. Nisbet et al., 2016), it is a region of major interest in the context of climate change and the associated risks. The Arctic is particularly sensitive to climate driven feedbacks. For instance, higher temperatures may favour methane production from wetlands and methane release from thawing permafrost as protected carbon becomes available to remineralization. This could drive a sustained carbon feedback to climate change (Schuur et al., 2015). Most major source types for methane are present in the northern high-latitudes: natural wetlands, oil and gas industry, and peat and forest burnings. There are also other sources that have received an increasing attention this past decade: freshwater systems (Walter et al., 2007; Bastviken et al., 2011; Tan and Zhuang, 2015; Wik et al., 2016), subsea permafrost and hydrates in the East Siberian Arctic Shelf (ESAS, in the Laptev and East Siberian Seas; Shakhova et al., 2010; Berchet et al., 2016; Thornton et al., 2016a) and terresrial thermokarst (Wik et al., 2016).

Methane sources and sinks can be estimated by a variety of approaches generally classified as either top-down (driven by atmospheric transport and concentration data) or bottom-up (driven by inventories and process-based models; e.g. Saunois et al., 2016). Our understanding of the methane global budget and its evolution is limited by the uncertainties about sources (their location, intensity, seasonality and proper classification) and sinks, by the representative coverage of the current observational surface network, by the biases of satellite-based data (e.g. Bousquet et al., 2018) and by the quality of atmospheric transport models (e.g. Patra et al., 2018). In particular, the discrepancies between bottom-up and top-down estimates remain a major concern both globally (Saunois et al., 2016) and in the Arctic (Thornton et al., 2016b; Thompson et al., 2017). Methane sources are particularly numerous, and temporally and spatially variable, especially when compared to carbon dioxide (Saunois et al., 2016). This makes it challenging to allocate emissions to each particular source as illustrated in Berchet et al. (2015), who studied overlapping wetland and anthropogenic emissions in Siberian lowlands with a top-down approach. Improving the attribution of methane emissions to specific processes in top-down approaches can benefit from the additional information (on top of the total concentrations) provided by the ratios of stable isotopes in atmospheric methane concentrations.

There are respectively three main stable isotopologues of methane that are commonly measured $^{12}CH_4$, $^{13}CH_4$ and $^{12}CH_3D$. Their respective abundances in the atmosphere are approximately 98.8%, 1.1% and 0.06% (Bernard, 2004). An isotopic signature characterizes each source and sink. The fractionation between the different isotopes is driven by source and sink processes that vary in space and time (Schwietzke et al., 2017). Microbial sources produce methane depleted in heavy isotopes. The isotopic signatures of biological sources vary depending on the metabolic pathway of formation, the nature of the degraded organic matter, on its stage of degradation, and on temperature (Whiticar, 1999). Thermogenic sources related to fossil fuels emit methane that tends to be not as depleted in heavy isotopes as microbial sources. Pyrogenic sources related to incomplete biomass combustion are even less depleted, with combustion of C3 plants contributing lighter signatures than C4 plants. Sink processes also influence methane's isotopic composition. The isotopic fractionations associated with the reaction with OH and the uptake by soils enrich atmospheric methane in heavier isotopes compared to the mean source signature. Atmospheric methane carries the isotopic signature resulting from the summed value of all of its sources and sinks. Though measurements of $^{12}CH_3D$ exist, only $^{12}CH_4$ and $^{13}CH_4$ are considered in this study because they are the most abundant methane isotopologues in the atmosphere and as such are easier to measure than $^{12}CH_3D$. Regular measurements using flask samples exist since the early 2000s for $^{13}CH_4$. Unfortunately $^{12}CH_3D$ flask measurement series are scarce, with no published Arctic series for recent years. Laser spectrometer-based instrument for $^{13}CH_4$ continuous measurements are currently being or have been settled at different locations (e.g., Zeppelin mountain, Svalbarg, since 2018), while it is less the case for $^{12}CH_3D$ likely because only one instrument is commercially available.

The isotopic variations are small: the ratio of $^{13}C/^{12}C$ in methane is expressed in conventional delta notation as $\delta^{13}C$-$CH_4$, which is the part per thousand deviation of the ratio in a sample to that in an international standard:

$$\delta^{13}C\text{-}CH_4 = [(R_{sample} / R_{standard}) - 1)] \times 1000 ‰ \qquad (1)$$

where R is $^{13}C/^{12}C$ of either the sample or of a community determined standard (currently Vienna-Pee Dee Belemnite, V-PDB; Craig, 1957).

The use of stable isotopes for discriminating methane sources is not new (Schoell, 1980). Isotope data can bring a valuable constraint on the methane budget (Mikaloff-Fletcher et al., 2004) and be relevant to eliminate different emission scenarios used to explain methane evolutions, globally (Monteil et al., 2011; Saunois et al., 2017) or regionally, for example in the Arctic (Warwick et al., 2016). Since 2007, globally averaged atmospheric methane concentrations have been steadily increasing and at the same time it has become more depleted in $^{13}C$. Nisbet et al. (2016) found the post-2007 shift in the $\delta^{13}C$-$CH_4$ value of the global atmospheric mean concentration to be -0.17‰. This shift signifies major ongoing changes in the methane budget and can be used to bring additional constraints on the source partitioning (Saunois et al., 2017). Using a box-model, Schaeffer et al. (2016) estimated the $\delta^{13}C$-$CH_4$ value of the post-2007 globally averaged source needed to match the observed $\delta^{13}C$-$CH_4$ evolution, to be -59‰. They concluded that the post-2007 rise was driven by microbial emissions, in particular from agricultural sources. The Schaeffer et al. (2016) estimate was used to validate the sectorial partition of the emission changes for 2000-2012 retrieved by Saunois et al. (2017). However, large uncertainties and overlaps remain for source signatures, implying that $\delta^{13}C$-$CH_4$ cannot points towards a unique solution.

Three main limitations remain in the use of isotopic data to improve our knowledge of methane sources and sinks: the wide ranges of isotopic signatures, the lack of information to estimate these signatures, and the lack of atmospheric isotopic data to assimilate in top-down approaches (Tans, 1997).

Isotopic signatures span large ranges of values, typical ranges being -70 to -55‰ for microbial, -55 to -25‰ for thermogenic and -25 to -13‰ for pyrogenic sources (Kirschke et al, 2013). Actually, significant overlap occurs (see Thornton et al., 2016b, and Section 2.4: e.g. -110 to -50‰ for microbial signatures, -80 to -17‰ for coalfields). Modelling studies do not always reflect these ranges because they choose only one or a few values for each source. McCalley et al. (2014) found that using the commonly used isotopic signature for wetlands for future emissions related to thawing permafrost could entail overestimations of a few $TgCH_4$ and an erroneous source apportionment. Regarding coal emissions, Zazzeri et al. (2016) pointed out that global models usually use a signature of -35‰ for coal, while measured values are between -30‰ and -60 ‰ depending on the coal type and depth (from anthracite to bituminous). Recently, Sherwood et al. (2017) compiled a global comprehensive database of $\delta^{13}C$-$CH_4$ and other methane isotopic signatures for fossil fuel, microbial and biomass burning sources. They pointed out that most modelling studies relied on a set of canonical isotopic signature values that circulated within the modelling community, which could have led to the use of erroneous values. For example, using a previous version of the Sherwood database, Schwietzke et al. (2016) revised the fossil fuel methane emissions upward by about 50% for the past three decades.

The lack of information on $\delta^{13}C$-$CH_4$ signatures is also a limitation for identifying sources of distinctive methane plumes (France et al., 2016). However, several recent measurement campaigns showed the value of determining $\delta^{13}C$-$CH_4$ for source apportionment. For example, Röckmann et al. (2016) have deployed high frequency isotopic measurements of both $\delta^{13}C$-$CH_4$ and $\delta D$-$CH_4$ at Cabauw in Europe and were able to identify specific events and to allocate them to specific anthropogenic sources (ruminants, natural gas or landfills). Similarly, the isotopic analyses led by Cain et al. (2016) from aircraft data in the North Sea made it possible to identify a source in a plume downwind of gas fields, which would have been missed without the isotopic information. In the Arctic, the importance of wetland emissions has been highlighted with the analysis of isotopic data from aircraft, ships and surface stations (Fisher et al., 2011; O'Shea et al., 2014; France et al., 2016). Field campaigns are also regularly organized to measure the isotopic signatures of various sources (Pisso et al., 2016; McCalley et al., 2014; Fisher et al., 2017).

The paucity of isotopic measurements to constrain top-down atmospheric inversions is another limitation. Inversions assimilating both total methane and isotope data are few; they use only flask sampling data, and rely on a few sites around the world. This, together with the lack of information on isotopic signatures can explain why such multi-constraint inversions have mostly been conducted with simple box-models so far (e.g. Schaefer et al., 2016). However, laser spectrometers can now provide continuous observations of methane isotopes with satisfying performance (Santoni et al., 2012). Moreover, such high frequency and high precision isotope measurements were shown, if applied to the current observational network, to potentially reduce uncertainties to source inversion in all sectors, even at the national scale (Rigby et al., 2012).

Even though no long-term continuous atmospheric $^{13}CH_4$ time series are yet available, it seems important to evaluate their potential to improve our knowledge on methane sources and sinks. A first step is the modelling of the isotopic signals to be expected at possible monitoring sites, taking into account the range of isotopic signatures of the different sources. The northern high-latitude region is chosen as a test region because of the significant potential of the climate-carbon feedback mentioned earlier and because methane emissions may overlap less (in time and space) than in the tropics for instance.

Following Thonat et al. (2017), who estimated the detectability of methane emissions at Arctic sites measuring total $CH_4$, this paper aims at extending this approach to $\delta^{13}C$-$CH_4$ observations, even if they do not exist yet. After presenting the 24 existing monitoring sites in the northern high-latitudes and the modelling framework (section 2), we evaluate how well our model simulates $\delta^{13}C$-$CH_4$ at the five sites where it is already monitored (section 3.1). Then, the atmospheric signals of the various northern high-latitude methane sources at these sites are estimated (section 3.2) before determining their detectability based on instrumental constraints and on the uncertainties of the isotopic signatures (section 3.3).

**2 Measurements and modelling framework**

2.1. Measurements

Measurements of the isotopic ratio in atmospheric methane for 2012 come from five northern high-latitude surface sites (White et al., 2018). The locations of these sites are shown in Fig. 1 and their characteristics are given in Table 1. Most of them are considered to be sampling background air: Alert is located in North Canada; Zeppelin (Ny-Ålesund) is on a mountaintop in the Svalbard archipelago; Cold Bay is in the Alaska Peninsula; and Summit is at the top of the Greenland Ice Sheet. The Barrow observatory, located in the North Slope of Alaska, is more affected by local wetland emissions. NOAA-Earth System Research Laboratory (NOAA-ESRL) is responsible for the collection and analysis of the weekly flask samples. The isotopic composition is determined by INSTAAR (Institute of Arctic and Alpine Research) of the University of Colorado. All data are reported in conventional delta notation, in per mil (‰). The $\delta^{13}$C-CH$_4$ observations are given with a precision of better than 0.1‰ (White et al., 2018). All data without reported issues in collection or analyses are selected; outliers above 3-sigma of the variability at the station are discarded.

Other sites where atmospheric methane is measured are also included in this study. They do not provide $\delta^{13}$C-CH$_4$ observations, but we evaluate their potential in doing so. Their description is given in Table 1 as well.

2.2 Model description

The Eulerian chemistry-transport model CHIMERE (Vautard et al., 2001; Menut et al., 2013) is used to simulate tropospheric $^{12}$CH$_4$ and $^{13}$CH$_4$ concentrations separately, the isotope ratio being computed offline a posteriori. Following Thonat et al. (2017), the domain has a regular kilometric resolution of 35 km, which avoids numerical issues due to too small grid cells close to the Pole encountered in regular latitude-longitude grids. It covers all longitudes above 64°N but extend partially to 39°N, as illustrated in Fig. 1. The troposphere is divided into 29 vertical levels from the surface to 300 hPa (~9000 m).

CHIMERE solves the advection-diffusion equation and is forced using meteorological fields from the ECMWF (European Centre for Medium Range Weather Forecasts, http://www.ecmwf.int/) forecasts and reanalyses. Wind, temperature, water vapour and other meteorological variables are given with a 3 h time resolution, at ~0.5° spatial resolution, and 70 vertical levels in the troposphere. Initial and boundary concentrations of $^{12}$CH$_4$ and $^{13}$CH$_4$ come from a global simulation of the general circulation model LMDZ (Hourdin et al., 2006) for the year 2012. This global simulation used emission fluxes (including ORCHIDEE for wetland emissions, EDGARv4.2 for anthropogenic emissions other than biomass burning and GFED4.1 for biomass burning emissions) that were adjusted in order to obtain a reasonable agreement at the global scale between the simulated isotopic signal and the flask measurements of the NOAA-ESRL network (Dlugockenky et al., 1994). These global fields have a 3 h time resolution and 3.75°x1.875° spatial resolution. These meteorological and concentration fields are interpolated in time and space within the grid of the CHIMERE domain.

The model is run with various tracers, each one corresponding either to the $^{12}$CH$_4$ or to the $^{13}$CH$_4$ component of a methane source. Simulated $^{12}$CH$_4$ and $^{13}$CH$_4$ of all sources are then used in the calculation of $\delta^{13}$C-CH$_4$. This allows us to analyse the contribution of each source in $\delta^{13}$C-CH$_4$. Three pairs of tracers correspond to anthropogenic sources: emissions from oil and gas; from solid fuels (coal); and other anthropogenic emissions (mostly from enteric fermentation and solid waste disposal). One pair of tracers corresponds to biomass burning. Two pairs correspond to geological sources: continental micro- and macro-seepages; and marine seepages. Three pairs correspond to other natural sources: wetlands, freshwater systems, and emissions from the ESAS. Another pair of tracers corresponds to soil uptake, considered as a negative surface source. Finally, one pair of tracers corresponds to the boundary conditions. No pair of tracers is implemented for the initial conditions: simulations in January are partly influenced by prescribed initial conditions from global fields during the spin up period of 2-4 weeks (typical mixing time of air masses in the domain with the chosen model set-up spanning high northern latitude regions) but this has little impact on our conclusions. No chemistry is included in the multi-tracers simulation, but another simulation is done including the reaction with OH in order to assess the contribution of this major sink. More details on the aforementioned emission categories are given below in Section 2.3.

2.3 Input emission data

Surface emissions used as inputs in the model come from various inventories, models, and data-driven studies. The emissions used are the same as in Thonat et al. (2017) where they are described and discussed in more details : we provide a summary below and in Table 2.

All anthropogenic emissions are taken from the EDGARv4.2FT2010 yearly product (Olivier and Janssens-Maenhout, 2012). When possible, the 2010 data are updated using FAO (Food and Agriculture Organization, http://www.fao.org/faostat/en/#data/) and BP (http://www.bp.com/) statistics (on enteric fermentation, and

manure management, and on oil and gas production, fugitive from solid, respectively). For 2012, anthropogenic emissions amount to 20.5 $TgCH_4$ $yr^{-1}$ in our domain, mostly from the fossil fuel industry. Biomass burning emissions come from the GFED4.1 (van der Werf et al., 2010; Giglio et al., 2013) daily product, and represent 3.1 $TgCH_4$ $yr^{-1}$ in our domain.


Wetland emissions are derived from the ORCHIDEE global vegetation model (Ringeval et al., 2010, 2011), on a monthly basis. Annual emissions from wetlands in our domain correspond to 29.5 $TgCH_4$ $yr^{-1}$. A large uncertainty affects wetland emissions, which can vary widely depending on the chosen land vegetation model and wetland area dynamics (e.g., Bohn et al., 2015). Emissions from geological sources stem from the GLOGOS database (Etiope, 2015), and amount to 4.0 $TgCH_4$ $yr^{-1}$ in our domain. ESAS emissions are prescribed to 2 $TgCH_4$ $yr^{-1}$, in agreement with the estimate made by Thornton et al. (2016) based on a ship measurement campaign, and with the estimate made by Berchet et al. (2016) based on atmospheric observations at surface stations. The temporal and geographic variability of the ESAS emissions is based on the description by Shakhova et al. (2010), following the modelling framework of Berchet et al. (2016).


Following Thonat et al. (2017), we consider that 15 $TgCH_4$ $yr^{-1}$ are emitted by all lakes and reservoirs located at latitudes above 50°N. The localisation of these freshwater systems relies on the GLWD level 3 map (Lehner and Döll, 2004). Our inventory was built based on some simplifications: the emissions are uniformly distributed among lakes and reservoirs; no emission occurs when the lake is frozen, and emissions are constant otherwise. Freeze-up and ice-out dates are estimated based on surface temperature data from ECMWF ERA-Interim reanalyses. Freshwater emissions amount to 9.3 $TgCH_4$ $yr^{-1}$ in our domain, which is consistent with recent pan-Arctic studies (e.g., Wik et al., 2016; Tan and Zhuang, 2015).


2.4 Source isotopic signatures

Source signatures are chosen constant in time and space in our modelling framework. Regional seasonal variations of microbial signatures are expected to be small (e.g. Sriskantharajah et al., 2012); some homogeneity can be assumed at the scale of our domain, which only comprises high northern latitudes; and possible heterogeneity is assumed to be smoothed out by the model 35 km horizontal resolution. Also, considering that most atmospheric sites are located far from large emission areas, the signals in the emissions are mixed by the atmospheric transport. Therefore, we have chosen to use only one value for each source but to test various scenarios with different isotopic signatures (see Sect. 3.2).


The Sherwood et al. (2017) data on fossil fuel emissions for countries within our domain show a wide range of measured isotopic signatures. For conventional gas and shale gas, data range between -76 and -24‰, with means, for Russia (number of data, n=556), Canada (n=490), Norway (n=28), and the US (Alaska) (n=20), of -46, -51, -44, and -43 ‰ respectively. Heavier signatures (typically -40‰) are generally used for oil and gas related emissions in global studies (e.g. Houweling et al., 2006; Lassey et al., 2007) and for Arctic studies as well (Warwick et al., 2016), but more depleted signatures have also been used for Russia (-50‰ in Levin et al., 1999). Given that Russia is by far the largest emitter of methane from natural gas production and distribution, we chose here a mean value of -46‰ for the whole domain, but test our results over a range spanning -40‰ to -50‰. As it is difficult to distinguish between methane associated to gas and oil exploitation, the same signature is used for both.




The range of isotopic values is also very large for emissions from coalfields: from -80 to -17‰ (Rice, 1993). Data are scarcer in the Sherwood et al. (2017) database than for natural gas, with just one reference for Russia and 92 reported values for Canada, the mean being -55‰. Russia is again the top emitter in this category, but the paucity of the data prevents us from using the single value for the whole domain. Zazzeri et al. (2016) highlighted the dependence of the isotopic value on the coal rank and type of mining, although national and regional specificities remain. Basically, the higher the coal rank (i.e. the carbon content), the heavier the isotopic signature. The main Russian coal basins, the Kuznetsk and Kansk-Achinsk basins, located in southern Siberia, where low rank coal is extracted, are not part of our domain. The few major hotspots of emission associated to coal in our domain, according to EDGARv4.2FT2020, correspond to basins where hard coal is exploited, and mainly bituminous coal (Podbaronova, 2010). According to the broad classification suggested by Zazzeri et al. (2016) for modellers, this means rather light isotopic signatures, between -55 and -65‰. Consequently, we chose here a mean value of -55‰ for emissions associated to coal in our domain, which is lighter than the values usually used in global methane budgets (e.g. -37‰ in Bousquet et al. (2006) and Tyler et al. (2007); -35‰ in Monteil et al. (2011)), but test our results over the range of -50‰ to -65‰.




Other non-negligible anthropogenic sectors in our domain are enteric fermentation and waste disposal. For the former, the $\delta^{13}C$ signature depends strongly on the ruminants' diet and on the species. Klevenhusen et al. (2010) found signatures from cows of -68‰ (C3 plants) or -57‰ (C4 plants), depending on the diet, in agreement with previous studies by Levin et al. (1993) and Bilek et al. (2001). Here, a value of -62‰ was used, as in other methane isotopic budgets (e.g. Tyler et al., 2007; Monteil et al., 2011). Methane emitted by organic waste is enriched as a result of methane oxidation after its production in the anoxic layer. Here, a value of -52‰ was used, in agreement with Chanton et al. (1999) (-58 to -49‰) and close to what was found by Bergamaschi et al. (1998b) (-55‰). The emissions of those two sources are an order of magnitude lower than anthropogenic emissions from fossil fuel production; thus, their isotopic signature does not significantly impacts the isotopic signal at observation sites .

Walter Anthony et al. (2012) found natural seeps concentrated along the boundaries of permafrost thaw and retreating glaciers in Alaska and Greenland, with a wide range of isotopic signatures, originating from fossil and also younger methane. However, geological methane is mostly of thermogenic origin (Etiope, 2009), and this is also true for submarine seepage (e.g. Brunskill et al., 2011). In this region, geological manifestations occur through submarine seepages and microseepages with mean isotopic signatures of about -51.2‰ and -51.4‰ with uncertainty in the order of 7‰ and 2‰, respectively (Etiope et al., 2019). As a consequence, the isotopic signature used here for geological methane, both continental and submarine, is -52‰, following Etiope et al. (2019), associated to the range -50‰ to -55‰.

The values of isotopic signatures for biomass burning are found in a small range, despite their dependency on the fuel type (C3 versus C4 plants) and the combustion efficiency. For example, Chanton et al. (2000) reported values comprised between -30‰ and -21‰ for US forests. Yamada et al. (2006) estimated the global biomass burning $\delta^{13}C-CH_4$ at -24‰, while Whiticar and Schaefer (2007) suggested -25‰. Here, the value of -24‰ was used, as a mean value, but signatures ranging from -30‰ to -21‰ have been tested (Table 3).

Microbial methane from wetlands has a wide range of isotopic signatures, varying from -110 to -50‰ (Whiticar, 1999). Acetoclastic fermentation results in methane relatively less depleted in $^{13}C$ ($\delta^{13}C-CH_4$ of -65 to -50‰), while $CO_2$ reduction produces methane highly depleted in $^{13}C$ ($\delta^{13}C-CH_4$ of -110 to -60‰) (Whiticar, 1999; McCalley et al. 2014). The partition between these two production pathways depends partly on the ecosystem type and season. The isotopic signature of the emitted methane also depends on other factors, such as the pathways of transport and oxidation (Chasar et al., 2000). Several studies on the isotopic signature of wetlands are compiled in Table 3, focusing on high northern latitudes. All studies report values generally ranging between -75‰ and -60‰. Here again, the difficulty in dealing with these reported source signatures has to do with their representativity. Some observations are from chamber studies, which, by nature, focus on very local signals; others are given by ambient air samplings and can be representative of several hundred square kilometres, so possibly encompassing other source and sink determinants. The chamber studies present a wide variety of values for the same site. For example, Fisher et al. (2017) reported values at the Stordalen Mire ranging from -112 to -48‰; even in the same week, changes can be as large as 30‰. The signals can also vary significantly with the time of year and the kind of ecosystem (McCalley et al., 2014). For example, for three different peatland systems in Finland, Galand et al. (2010) report values that differed by 30‰. Consequently, values in Table 3 are mostly derived from ambient air samplings rather than chamber measurements, and we give means rather than the whole measured ranges. The value of -70‰ was used in our study, close to the recommendation to modellers made by Fisher et al. (2017) (-71 ± 1‰) and France et al. (2016) for wetlands above 60°N. However we tested a wide range of signature for wetland emissions between -80 and -50‰.

Most values labelled "Wetlands" in Table 3 encompass not only wetlands but also a mix of wetlands and other exposed freshwater systems.  Shallow lakes, ponds and pools, common in the Arctic, have not always been considered a distinct source (Bastviken et al., 2011). This is another limitation in estimating the global methane budget (Saunois et al., 2016). Signature estimates based on air sampling are representative of a wide area, where exposed freshwaters are undoubtedly present. Moreover, signature ranges reported specifically from Arctic lakes are not precise enough to distinguish between water body types, and overlap those of wetlands (Wik, 2016). In the range of recent reported values (Walter et al., 2008; Brosius et al., 2012; Bouchard et al., 2015; Wik, 2016; Thompson et al., 2016), and close to the value used for Arctic wetlands, the value of -66‰ was used for the isotopic signature of freshwater system (here lakes and reservoirs) emissions in our domain. We also tested a wide range of signature for freshwater emissions between -80 and -55‰.

Sources of methane in the ESAS are varied and it is still a challenge to determine the origin of methane produced and emitted there (Ruppel, 2015). The shallow ESAS is underlain by formerly subaerial permafrost that has been

flooded by sea level rise since the Pleistocene (Dmitrenko et al., 2011). Carbon can be released via the degradation of permafrost or decomposition of gas hydrates. Sapart et al. (2017) showed that sediments in ESAS have isotopic signatures ranging between the two main microbial methane formation pathways. In an earlier study, Cramer and Franke (2005) observed significantly heavier $CH_4$ ($\delta^{13}$C-CH$_4$ ~-39.9‰) in Laptev Sea near-surface sediments, attributed to a deep thermogenic source. A wider range, with much lighter $CH_4$ was detected in the Laptev seawater column. Methane in the water is more enriched in $^{13}$C than in sediments, but the signature of methane emitted in the atmosphere is in the range of wetland emissions. Based on fewer data than Sapart et al. (2017), Overduin et al. (2015) reported more positive values, associated to strong $^{13}$C enrichment in the upper thawed permafrost layers. Oxidation in marine systems can be coupled to sulfate reduction as well in sub-oxic environments. This will not affect the atmospheric values directly but will shift the source signatures of the methane that is emitted from the surface to heavier values after having been diffusively advected from its sedimentary sites of production through the water column to the atmosphere. A mean signature of -58‰ (range -80 to -50 ‰) was used here for emissions from ESAS, in the range of the literature (Etiope et al., 2019).

2.5 Sinks: isotopic fractionation

The main sinks of methane in the troposphere are its oxidation by hydroxyl radicals (OH), which accounts for about 90% of the total sink (Saunois et al., 2016), its reaction with chlorine (Cl) in the marine boundary layer (about 3%) and its uptake by soils (about 3%, at the global scale; Kirshke et al., 2013).
Due to the difference in mass between the $^{12}$CH$_4$ and $^{13}$CH$_4$ isotopologues, chemical reactions in the atmosphere preferentially consume the lighter isotopologue, potentially causing significant fractionation. This is one of the reasons why the $\delta^{13}$C of methane in the atmosphere is not the same as that of the total source.
The chlorine sink is not included in our regional simulation. We have shown in Thonat et al. (2017) that this sink has a negligible impact of $CH_4$ mixing ratio (below 1 ppb in our polar domain).
Methane uptake occurs in unsaturated oxic soils due to the presence of methanotrophic bacteria. This sink may be particularly important in high latitude regions with wetlands. In our domain of simulation, its magnitude is equal to -3.1 Tg CH$_4$ yr$^{-1}$ (see Table 2).

Sinks can be characterised by their kinetic isotope effect (KIE), the ratio of the reaction rate coefficients (k) for two different isotopologues of the same molecule: $k_{light}/k_{heavy}$. For the reaction with OH this value is 1.0039 (Saueressig et al., 2001). For the soil uptake, the KIE is 1.020, which is represented by a fixed $\delta^{13}$C-CH$_4$ source signature of -65.7‰ in our model set-up. Despite a high KIE, including the chlorine sink in the regional simulation will not change significantly our conclusions on the local source detectability.

**3 Results**

Simulations of distinct tracers, each one corresponding to a different $^{12}$CH$_4$ or $^{13}$CH$_4$ source, are run with CHIMERE for the year 2012. Since isotopic signatures generally vary over a wide range for a given source (Sect. 2.3), we ran simulations using the mean value and the extreme values of the range given in Table 2 for oil and gas, coal, biomass burning, wetland, freshwater, and ESAS emissions.

3.1 Comparison between modelled and observed $\delta^{13}$C-CH$_4$

Most of the five sites where weekly $\delta^{13}$C-CH$_4$ measurements are available are remote from any emitting areas (Fig. 1), with the exception of Barrow where significant methane enhancements from nearby wetlands can happen in summer (Sweeney et al., 2016). The boundary conditions are the dominant signal in our domain, especially in winter, both in terms of total methane mixing ratio (in ppb) and $\delta^{13}$C-CH$_4$ value (in ‰), as illustrated in Fig. 2. The boundary conditions represent methane coming from lower latitudes south of the polar domain (Fig. 1). However, they cannot be fully considered as a background level of methane given that (i) they may be due to emissions from the northern high latitudes that have left our domain and then re-entered it; (ii) they may bring to the domain air masses that are particularly depleted or enriched in methane.

For most remote sites, the maximum $\delta^{13}$C-CH$_4$ is reached in May-June and ranges between -47.3 and -47.1‰ (Fig. 2). Then wetlands and freshwater systems start emitting $^{13}$C-depleted methane and the minimum is reached in September-early November, with values around -47.8‰. One exception is Cold Bay where $\delta^{13}$C-CH$_4$ in January was much lower than other sites. In Barrow, the minimum reaches -48.2‰. The yearly mean is -47.6‰ at Barrow and -47.5‰ at the other sites. The seasonal amplitude is about 0.6‰. The variability of the measurements is higher in Barrow and Cold Bay compared to the three others, highlighting that these two sites are the most sensitive to northern high latitude sources (mainly wetland emissions) at the synoptic scale.

The contribution of the boundary conditions to simulated $\delta^{13}$C-CH$_4$ is approximately between -47.2 and -47.6‰. The increment added by northern high latitude sources lies between -0.1 and -0.2‰ in summer (June-October), except in Barrow where it is -0.4‰, and is close to zero in winter (November-May). Barrow is more sensitive to the regional sources (mainly wetland and freshwater emissions) compared to the four other sites (see Fig S4 against Fig 4, S1, S10 and S18). On a yearly basis, our model overestimates $\delta^{13}$C-CH$_4$. The large overestimation in winter (~0.2‰) is due to the boundary conditions that are too high in terms of total methane compared to continuous measurements (as shown in Thonat et al., 2017). Too large contributions of low latitude fossil sources lead to too high $\delta^{13}$C-CH$_4$ values. Nevertheless, large spikes are simulated in winter at Barrow and Alert, some of which are attributed to ESAS emissions. Due to the low frequency of flask measurements, it is not possible to associate these simulated spikes to observed ones. Higher frequency measurements are needed to assess the reality of such spikes and their magnitudes, and to allow discussion on both the magnitude of the source(s) and its/their isotopic signature(s). In summer, the model underestimates $\delta^{13}$C-CH$_4$ by less than 0.11‰ at all sites, which is in the range of the uncertainty of the measurements. However, the seasonality is only fairly captured by the model. The decrease in early summer comes too soon and so does the autumn minimum, as already noticed by Warwick et al. (2016). Thonat et al. (2017) demonstrated that this result is mostly emission-driven: the seasonality of wetland emissions is not well reproduced by the various existing land surface models because wetland emissions derived from biogeochemical models occur too soon and cover too short a period during the year.

Despite their importance to assess the inter-annual variability and seasonality of $\delta^{13}$C-CH$_4$, the available flask measurements do not allow us to quantify the ability of the model to represent the synoptic variations. Continuous measurements of $\delta^{13}$C-CH$_4$, as well as $\delta$D-CH$_4$, would be necessary to evaluate the model in a more quantitative way. Even though further improvements will be necessary in the model, we assume in the following that the model performances associated to sensitivity tests using various isotopic signatures are sufficient for estimating the magnitude of the isotopic signals in $\delta^{13}$C-CH$_4$ originating from the various northern latitude sources.

3.2 Contributions of northern high-latitude sources in $\delta^{13}$C-CH$_4$ at northern latitude sites

In terms of total methane, our domain is dominated by anthropogenic sources in winter, and by wetland emissions in summer. ESAS and geological sources can also have a relatively significant impact in winter in some areas, while freshwater systems are an important contributor to atmospheric methane in summer (Thonat et al., 2017). The spatial distribution of the source contribution to the $\delta^{13}$C-CH$_4$ value depends on the magnitude of the emission but also on the difference between the isotopic signature of the source and of the boundary conditions. The difference between total $\delta^{13}$C-CH$_4$ and the contribution of the boundary conditions (Figure 2, black and cyan lines, respectively) represents the sum of the direct contribution from the various northern latitude sources at the measurement locations. The combination of the various signals due to northern latitude sources depends on the station, as shown in Fig. 2.

These five sites do not form a large-enough sample to be representative of all northern latitude sites. Therefore, Figure 3 shows the winter and summer means of the simulated direct contributions of the various sources to the $\delta^{13}$C-CH$_4$ value at the 24 sites of Fig. 1. For each site, the seasonal mean contribution of each source is plotted along a cumulative dotted line. The rightmost black point of each line represents the total contribution of all northern latitude sources i.e. the difference between simulated total $\delta^{13}$C-CH$_4$ and $\delta^{13}$C-CH$_4$ from the boundary conditions alone. The frequency distribution of the contribution from all the Arctic sources to the signal is over plotted with an arbitrary unit, showing the range of isotopic signal covered over the season. For example, if we consider Tiksi (TIK) in winter: the direct contribution of all Arctic sources is -0.09‰ on average over the season. However, the frequency distribution shows that the isotopic contribution at Tiksi is mainly between 0 and -0.2‰ but can reach lower values up to -0.25‰.

On average, the contributions of northern high-latitude sources to the isotope ratio are very low in winter at all sites, between -0.65 and +0.03‰. The isotope ratio signal is low in winter because the largest contribution of Arctic sources to atmospheric methane in this season is due to oil and gas emissions, whose signature (-46‰) is very close to that of boundary conditions. One exception is YAK, where the mean winter contribution to $\delta^{13}$C-CH$_4$ is -0.63‰. This is due to large simulated mixing ratios of methane from nearby coal emissions. The daily isotope ratio signal shift due to Arctic contributions there can reach -1.75‰. Geological emissions have a signature close to oil and gas in our modelling framework and do not show up in the simulated signal. On the contrary, ESAS emissions have an impact on $\delta^{13}$C-CH$_4$ at some sites at the synoptic scale: the maximum $\delta^{13}$C-CH$_4$ northern high-latitude contribution at AMB and CHS in winter is ~-0.5‰, and ~-0.4‰ at TIK, which are close to the shores of ESAS. NOY is the only site with a positive mean contribution to $\delta^{13}$C-CH$_4$ in winter. Large

enhancements of $^{12}CH_4$ from oil and gas, which in NOY regularly exceeds 100 ppb in winter, succeed in making a significant difference with the $\delta^{13}C$-$CH_4$ value of the boundary conditions. Apart from NOY, the northern high-latitude contribution to $\delta^{13}C$-$CH_4$ is very rarely positive among the sites, and stays low when it is positive (maximum is 0.13‰ at DEM).


Compared to winter, higher contributions of northern high-latitude sources to the $\delta^{13}C$-$CH_4$ values are found in summer at most stations because of the large magnitude of natural emissions, especially from wetlands. Wetland emissions contribute more than two third of the signal at all sites, except BKL and CBB where the contribution of freshwater systems is also important, and YAK (again due to coal emissions). Wetlands keep the isotope ratio

quite low, with two sites having a mean $\delta^{13}C$-$CH_4$ contribution more negative than -1.0‰ (BCK, INU). Values below -2.0‰ are even reached on a daily basis at 15 sites; it is frequent at BCK for example, where the influence of wetlands and freshwater systems are combined. On top of wetland and freshwater influences, ESAS explains more than 10% of the signal at TIK and AMB.

Figure 3 reveals what can be expected on a seasonal basis at the different sites, but does not show how the various source contributions combine all along the year and how different source signatures can affect the total $\delta^{13}C$-$CH_4$ signal. Figure 4 and the supplementary Figures S1-S23 show the time series of the direct contribution of each source and sink to the total $\delta^{13}C$-$CH_4$ at the 24 northern latitude stations. A focus is put on Zeppelin station with Fig. 4 because a new Aerodyne instrument has been installed there during Summer 2018 to

continuously measure $\delta^{13}C$-$CH_4$ for at least one year. Figure 4 illustrates the magnitude and timing of the maximum signal of each source during the year, the potential compensation between sources, and the seasonality of the various contributions.

Zeppelin is a typical example of a remote site. $\delta^{13}C$-$CH_4$ from anthropogenic emissions are very small (<0.02‰,

except some particular events when considering the lightest isotopic signatures) because the source areas are far from the station, and anyway tend to cancel out because the signals from oil and gas, and from coal have approximately the same magnitude, but opposite signs. The signal from geological sources remains negligible being one order of magnitude lower than anthropogenic sources. Only wetland emissions succeed to tear the signal away from the value of the boundary conditions, from June to October, with synoptic changes up to -

0.2‰. Freshwater systems intensify the signal by 0.02‰ on average in summer, with maxima around 0.05‰ on a synoptic basis. These contributions are diminished by biomass burning (~+0.01‰) and also by the fractionating effects of the two major sinks (~+0.01‰). The simulated $\delta^{13}C$-$CH_4$ signal at the site is the result of these competing signals. Varying the isotopic signatures of natural sources does not change the conclusions with wetland, freshwater and ESAS synoptic events reaching at maximum respectively -0.3‰, -0.1‰ and -0.15‰.

Therefore, in the case of a remote station such as ZEP, signals of individual sources remain below 0.3‰ at the synoptic scale and partial compensation between sources determines the total $\delta^{13}C$-$CH_4$ anomaly.

Analysing other stations (Figures S1 to S23) reveals that synoptic events larger than 2‰ due to summer wetland emissions could happen at AMB, BCK, CHK, DEM, IGR, INU, NOY, TIK. For freshwater emissions, events

larger than 0.5‰ are simulated at AMB, BKL, BRW, BCK, CBB, CHU, and INU. For ESAS, varying the isotopic signature induces synoptic events larger than 0.3‰ at some sites (AMB, BRW, CHS, TIK). When varying the isotopic signature of anthropogenic emissions, DEM, IGR, KRS, NOY and VGN show synoptic events due to oil and gas that are larger than 0.15‰, and only YAK shows synoptic events due to fugitive emissions larger than 1‰; these events occur mainly in winter. Biomass burning synoptic events are the largest

at BCK, DEM, KRS, NOY, and YAK with events larger than 0.2‰.

The influence of the sinks on synoptic variations remains smaller than 0.05‰ at most sites. Note that the sink constituted by the reaction with Cl radicals in the marine boundary layer is not taken into account here, given its very small impact on $CH_4$ mixing ratios in our domain (less than 1 ppb, Thonat et al., 2017), although it is highly

fractionating. As aforementioned, including this sink in the regional simulation will not change significantly our conclusions on the local source detectability.

3.3 Detectability of northern high-latitude sources using isotopic measurements

The magnitude of $\delta^{13}C$-$CH_4$ signals to be expected at present and potential measurement sites and the contributions of individual sources to these signals do not lead directly to quantifying the detectability of individual sources, as the latter also depends on the performances of the measuring instrument. Here we focus on a detectability definition taken from a regional inversion point of view: regional inversion systems analyse daily signals and optimize sources depending on synoptic deviations of the observed signals compared to the

simulated ones. Therefore, a measuring instrument is considered to provide useful information to the inversion

only if the synoptic variability of the atmospheric signal can be detected. To that end, we compute detectability capability in Fig. 5 and Tab. 4 as follows: (1) we compute the standard deviation over a five-day running window of the simulated total isotopic signal; (2) for a set of instrument precision threshold (from 0.2 to 0.01‰ see Fig. 5 and Tab. 4), if the running standard deviation is higher than the corresponding threshold, the source with the higher running standard deviation for the same 5-day window is considered detected for that one day; (3) for each threshold, we count the number of days over the year that each source is detected. Although the total atmospheric signal integrates contributions from different sources with different isotopic signatures, we keep only the major source contributing to the signal as a first order signal.

The range of instrument precision threshold was chosen according to present isotopic instrument systems. The flask measurements used in Sect. 3.1 (Tab. 1, Fig. 1 and 2) have an uncertainty of about 0.1‰. They are obtained using GC-IRMS (gas chromatography isotope ratio mass spectrometry; White et al., 2018). Using continuous-flow isotope ratio mass spectrometry, Fisher et al. (2006) reached a precision of 0.05 ‰. Laser-based instruments, using Cavity Ring Down Spectrometry or direct absorption spectrometry (Nelson et al., 2004) have been developed for 10 years for $CO_2$ isotopes (McManus et al., 2010) and, more recently for methane (Santoni et al 2012). The Aerodyne QCL instrument has proven to be capable of high frequency ($\geq 1$ Hz) measurements of $^{12}CH_4$ and $^{13}CH_4$ isotopes of $CH_4$ with *in situ* 1 second RMS $\delta^{13}C_{CH_4}$ precision of 1.5‰ and an Allan-minimum precision of 0.2‰ at 100 seconds (Santoni et al., 2012), recently improved to 0.1‰ through laser stability improvements. Such a small value (0.1 ‰) reaches the precisions reported for GC-IRMS (0.1‰). However, Aerodyne instruments face a strong drift that imposes a strict calibration protocol (every 2 hours in most recent set-ups), which dramatically reduces the daily number of available observations to typically a few tens. Depending on our capability to deploy stable and well calibrated instruments in very remote high latitudes sites, state-of-the art isotopic instruments may provide from a few to hundreds of independent data points per day, thus potentially improving the instrument precision of daily averaged observations up to 0.01‰

Detectability thresholds at the 24 sites of Table 1 are summarized in Table 4 and Fig. 5 when considering the mean values of the isotopic signatures of Table 2. Results for a 0.5‰ threshold is not shown in Fig. 5 because only YAK can detect sources (only the oil and gas sector) at this level of instrument precision. At ZEP, with an uncertainty higher or equal to 0.1‰, no source is detected. Currently, daily flask are operated at ZEP with an uncertainty of 0.05‰ but contamination problems occur. If such contaminations are avoided so that the measurement uncertainty reaches 0.05‰, some wetland events may be detected during about 10 days. From 0.05‰ of measurement uncertainty, the number of events is larger and other sources (freshwater and ESAS emissions) might be detected. At only 0.01‰, there were about 20 days of possible detection for ESAS a few days for freshwaters and none for anthropogenic emissions. Looking at results for all stations, wetland emissions are the most easily detected with more than 50 days for a measurement uncertainty above 0.1‰ for most sites (exception of ALT, BKL, CHL, ICE, PAL, SUM, SUM, ZEP, ZOT); the best scores of detection, with more than 150 days, are achieved at BCK, INU, DEM, and NOY. Freshwater emissions are easiest to detect at BKL and CBB with 100 days and 50 days above 0.1‰ respectively. Anthropogenic emissions are easily detected at YAK due to its close location to coal extraction sites. With a 0.05‰ uncertainty, most of the stations offer opportunities to detect regional sources, except remote stations and/or stations close to the boundaries of our domain (ALT, CHL, ICE, SUM, ZEP). For ESAS emissions, the minimum detection ranges between 0.02‰ and 0.1‰ depending on stations. ESAS emissions are best detected at AMB, CHS, and TIK with more than 50 days above 0.05‰. A few other sites offer detectability if uncertainties are lower than 0.02‰ (ALT, BRW, BKL, CBB, CHL, INU, and ZOT). As already noticed, the effect of anthropogenic emissions dominates at YAK with about 100 days above 0.2‰ uncertainty. Other sites located in Russia are able to detect anthropogenic emissions with more than 50 days of events above 0.02‰ (DEM, IGR, NOY, and VGN). Excluding YAK, the minimum detection of anthropogenic emissions ranges between <0.01‰ and 0.05‰ depending on stations. For the year 2012, YAK and KRS detect some biomass burning events with an uncertainty lower than 0.2‰ and 0.1‰, respectively. Geological sources are detected at ZOT when the uncertainty is lower than 0.01‰.

## 4 Discussion & conclusion

Although no continuous $\delta^{13}C$-$CH_4$ observed time-series are available yet, inverse modelers have been considering $\delta^{13}C$-$CH_4$ observations as promising to distinguish methane sources for a while (e.g. Hein et al. 1997). The assimilation of $\delta^{13}C$-$CH_4$ flask data into 3D-chemistry-transport global models has shown small changes in the balance of sources, involving mostly biomass burning at the global scale (Bousquet et al., 2006, see their supplementary page 7). This modest impact was explained by the scarcity of $\delta^{13}C$-$CH_4$ observations (only 13 flask stations in Bousquet et al., 2006), and the uncertainties on isotopic signatures. Since then the former has slightly improved at the global scale (20 flask sites reported in the World Data Center for Greenhouse Gases database at present; gaw.kishou.go.jp/) and continuous measurements are expected (e.g. Thornton et al.,

2016b) but the latter is still an issue because it is necessary to obtain precise isotopic signatures at the regional scale for the various processes emitting methane. 3D atmospheric forward modeling has also been used to interpret methane changes of the past decades through scenarios of methane emissions, methane sinks, and isotopic signatures (Monteil et al., 2011; Warwick et al., 2016), demonstrating the added-value of the global

monitoring of methane isotopes, although the above limitations are still present. Taking into account these limitations, most recent inverse studies integrating $\delta^{13}$C-CH$_4$ data have only used simple box-models and, therefore, have assimilated hemispheric or global mean time-series of $^{13}$C observations (e.g., Schaefer et al., 2016, Turner et al., 2017; Schwieztke et al., 2016). Such studies use strong simplifications in their setup and can obviously only address hemispheric to global scale emissions and trends.


Our work aims at preparing 3D inversions assimilating future continuous $\delta^{13}$C-CH$_4$ time-series to address the reduction of uncertainties on methane emissions at the regional scale. The northern high-latitudes have been chosen to make this first analysis because it is a climate-sensitive region (with potentially larger methane sources than today in the context of a changing climate) and because the mix of methane sources is less complicated than

in the tropics. Even in this apparently favorable context, the situation of the detectability of methane sources using $\delta^{13}$C-CH$_4$ observations is found challenging for at least three reasons. First, as already noted in Thonat et al. (2017), most of the methane signals received at northern latitude stations at the synoptic to seasonal scales come from lower latitudes, thus limiting the expected signal to noise ratio of the northern high-latitude sources. Second, the analysis presented in Sect. 3 reveals that, if isotopic signals from wetland emissions should be

detectable at most existing sites with reasonable measurement uncertainties on a daily basis (~0.15‰), detecting other sources would require more challenging measurement uncertainties: typically less than 0.05‰ for freshwaters, ESAS, and anthropogenic emissions (except at YAK); and less than 0.02‰ for other sources. Such ambitious values require solving or at least monitoring precisely the present drifts of existing instruments and stress the importance of having a precise scale for regular calibration. Third, the vision per source developed

here is optimistic as total isotopic signals received at stations may cancel each other out for some events, thus reducing the number of useful events constraining individual sources. It should be noted that we provide here a first order contribution in the signal, while air is mixed in the atmosphere and the total signal integrates contributions from different sources. As a result, the threshold and the main contributing source both depend on the isotopic signatures assigned to the different sources (Supplementary Fig. S24 to S27). For example, if the

heaviest (-50‰) isotopic signature from Table 2 is assigned to wetland emissions, then this source is hardly detected for measurement uncertainties higher than 0.05‰, while the lightest signature allows a detection for a 0.2‰ measurement uncertainty at more than half the sites. Similarly, freshwater or ESAS emissions are considered detectable with a measurement uncertainty of 0.2‰ at Russian sites when applying the lightest isotopic signatures. This study has been carried out only for the year 2012 as a test case. However, not all

emissions have a high inter annual variability, as does biomass burning. As a result, our findings should be valid for the other sources for most of the years over a few future decades.

Next steps of this work involve i) the analysis of more than one year of continuous measurements of $\delta^{13}$C-CH$_4$ at ZEP, ii) the refinement of isotopic signatures of the various emissions at the regional scale, iii) the

implementation of $\delta^{13}$C-CH$_4$ in inversion schemes in order to estimate the potential (if only pseudo continuous data were available) or the actual impact of $\delta^{13}$C-CH$_4$ in improving the estimation of regional methane emissions by 3D atmospheric inversions, and iv) assessing the potential of $\delta$D-CH$_4$ in both global and regional modelling framework.

**Data availability**
The data for $\delta^{13}$C-CH$_4$ observations were downloaded from the world Data Center for Greenhouse Gases (WDCGG) at https://gaw.kishou.go.jp. Datasets for the input emissions were downloaded from EDGAR and GFED databases. The modelling output files are available upon request to the corresponding author.

**Author contribution**
Thibaud Thonat, Marielle Saunois, Philippe Bousquet and Isabelle Pison designed the study. Brett F. Thorton and Patrick M. Crill brought expertise on observation availability and instrument performance. Thibaud Thonat performed the regional simulations. Thomas Hocking performed the global simulations used as boundary conditions. Antoine Berchet and Thibaud Thonat produced the figures. Thibaud Thonat and Marielle Saunois

prepared the manuscript. All co-authors contributed the analysis, the design of the figures, and the text.

**Competing interests.**
The authors declare that they have no conflict of interest.

**Acknowledgements.**

The authors acknowledge the principal investigators, Bruce Vaughn, James White and Sylvia Michel, of the five observation sites measuring $^{13}CH_4$ in the Arctic regions, whose data were used in this study, for maintaining methane measurements at high latitudes and sharing their data through the World Data Center for Greenhouse Gases (WDCGG). This work has been supported by the Franco-Swedish IZOMET-FS "Distinguishing Arctic
CH$_4$ sources to the atmosphere using inverse analysis of high-frequency CH$_4$, $^{13}$CH$_4$ and CH$_3$D measurements" project. The study extensively relies on the meteorological data provided by the ECMWF. Calculations were performed using the computing resources of LSCE, maintained by François Marabelle and the LSCE IT team. The authors warmly acknowledge the two anonymous referees whose help improved the manuscript and the conclusions of this study.

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

**Table 1.** Description of the 24 sites measuring methane used in this study and included in our polar domain.

| Code | Sites | Coordinates | Altitudes (m a.s.l) | $\delta^{13}$C-CH$_4$ observations |
|---|---|---|---|---|
| ALT | Alert | 82.45°N, 62.52°W | 36 | Y |
| AMB | Ambarchik | 69.62°N, 162.30°E | 5 | - |
| BKL | Baker Lake | 64.17°N, 95.50°W | 10 | - |
| BRW | Barrow | 71.32°N, 156.60°W | 2 | Y |
| BCK | Behchoko | 62.80°N, 116.10°W | 179 | - |
| CBB | Cambridge Bay | 69.10°N, 105.10°W | 30 | - |
| CAR | CARVE Tower | 65.00°N, 147.60°W | 611 | - |
| CHS | Cherskii | 68.61°N, 161.34°E | 23 | - |
| CHL | Churchill | 58.75°N, 94.07°W | 9 | - |
| CBA | Cold Bay | 55.21°N, 162.72°W | 25 | Y |
| DEM | Demyanskoe | 59.79°N, 70.87°E | 71 | - |
| IGR | Igrim | 63.19°N, 64.42°E | 53 | - |
| INU | Inuvik | 68.30°N, 133.50°E | 10 | - |
| KRS | Karasevoe | 58.25°N, 82.42°E | 78 | - |
| NOY | Noyabrsk | 63.43°N, 75.78°E | 100 | - |
| PAL | Pallas | 67.97°N, 24.12°E | 301 | - |
| ICE | Storhofdi | 63.40°N, 20.29°W | 118 | - |
| SUM | Summit | 72.60°N, 38.42°W | 3178 | Y |
| TER | Teriberka | 69.20°N, 35.10°E | 83 | - |
| TIK | Tiksi | 71.59°N, 128.92°E | 123 | - |
| VGN | Vaganovo | 54.50°N, 62.32°E | 197 | - |
| YAK | Yakutsk | 62.09°N, 129.36°E | 198 | - |
| ZEP | Zeppelin | 78.91°N, 11.89°E | 126 | Y |
| ZOT | Zottino | 60.80°N, 89.35°E | 104 | - |

**Table 2. Methane emissions and isotopic signatures in the studied domain (see text, Sect. 2.3 and 2.4). Emission and sink fluxes used here are the same as in Thonat et al. (2017).**

| Type of source/sink | Emissions (TgCH$_4$ yr$^{-1}$) | $\delta^{13}$C-CH$_4$ (‰) / KIE | Range $\delta^{13}$C-CH$_4$ (‰) |
|---|---|---|---|
| Oil and gas | 11.9 | -46 | -40,-50 |
| Coal mining | 4.7 | -55 | -50,-65 |
| Animals | 1.3 | -62 | − |
| Landfills | 1.1 | -52 | − |
| Total anthropogenic | 20.5 | | − |
| Biomass burning | 3.1 | -24 | −21,-30 |
| Geology | 4.0 | -52 | − |
| ESAS | 2.0 | -58 | -80,-50 |
| Wetlands | 29.5 | -70 | -80, -55 |
| Freshwater systems | 9.3 | -66 | -80, -50 |
| Soil uptake | -3.1 | -65.7 / 1.020 | − |
| OH oxidation | − | 1.039 | − |

**Table 3. $\delta^{13}C$-$CH_4$ source signatures reported for wetlands at high northern latitudes.**

| Measurements location | Type of source | Reference | $\delta^{13}C$-$CH_4$ (‰) |
|---|---|---|---|
| Manitoba, Canada | Tundra | Wahlen et al. (1989) | -62.9 |
| Ontario, Canada | Wetlands | Kuhlman et al. (1998) | -60.0 |
| Ontario, Canada | Wetlands | Fisher et al. (2017) | -67.2 |
| Saskatchewan, Canada | Wetlands | Fisher et al. (2017) | -66.8 |
| Alberta, Canada | Wetlands | Popp et al. (1999) | -66.3 to -63.6 |
| Alaska, USA | Tundra | Quay et al. (1988) | -64 |
| Alaska, USA | Wetlands | Martens et al. (1992) | -65.8 |
| Siberia, Russia | Wetlands | Meth-MonitEUr (2005) | -67.1 |
| Siberia, Russia | Wetlands | Tarasova et al. (2006) | -62.8 |
| Siberia, Russia | Wetlands | Bergamaschi et al. (1998) | -62.4 |
| Siberia, Russia | Wetlands | Sugawara et al. (1996) | -75 to -67 |
| Siberia, Russia | Wetlands (thermokarst basins) | Nakagawa et al. (2002) | -61.1 |
| Northern Fennoscandia | Wetlands | Fisher et al. (2017) | -72.0 to -69.2 |
| Lompolojänkkä, Finland | Wetlands | Sriskantharajah et al. (2012) | -68.7 to -64.9 |


**Table 4. Minimum detectability threshold (in ‰) of high northern latitude sources at all observation sites in 2012 considering the mean values of isotopic signature in Table 2. See Sect. 3.3 for the definition of the detectability threshold.**


| Station | Anthropogenic | Geology | Biomass burning | Wetlands | Fresh-waters | ESAS |
|---|---|---|---|---|---|---|
| ALT | - | - | - | 0.05 | - | 0.02 |
| AMB | - | - | - | 0.5 | - | 0.1 |
| BKL | - | - | - | 0.2 | 0.2 | 0.01 |
| BRW | - | - | - | 0.2 | 0.1 | 0.02 |
| BCK | - | - | - | 0.5 | 0.15 | - |
| CBB | - | - | - | 0.2 | 0.1 | 0.01 |
| CAR | - | - | - | 0.2 | - | 0.01 |
| CHS | - | - | - | 0.5 | - | 0.05 |
| CHL | - | - | - | 0.2 | - | 0.01 |
| CBA | - | - | - | 0.15 | - | 0.01 |
| DEM | 0.02 | - | - | 0.2 | - | - |
| IGR | 0.02 | - | - | 0.2 | 0.02 | - |
| INU | - | - | - | 0.5 | - | 0.01 |
| KRS | 0.01 | - | - | 0.2 | - | - |
| NOY | 0.05 | - | - | 0.2 | - | - |
| PAL | - | - | - | 0.05 | 0.05 | - |
| ICE | - | - | - | 0.05 | 0.01 | - |
| SUM | - | - | - | 0.02 | - | - |
| TER | 0.02 | - | - | 0.1 | 0.02 | - |
| TIK | - | - | - | 0.2 | - | 0.05 |
| VGN | 0.02 | - | - | 0.2 | 0.02 | - |
| YAK | 0.2 | - | 0.1 | 0.15 | - | - |
| ZEP | - | - | - | 0.02 | 0.05 | 0.01 |
| ZOT | - | - | - | 0.05 | - | 0.05 |

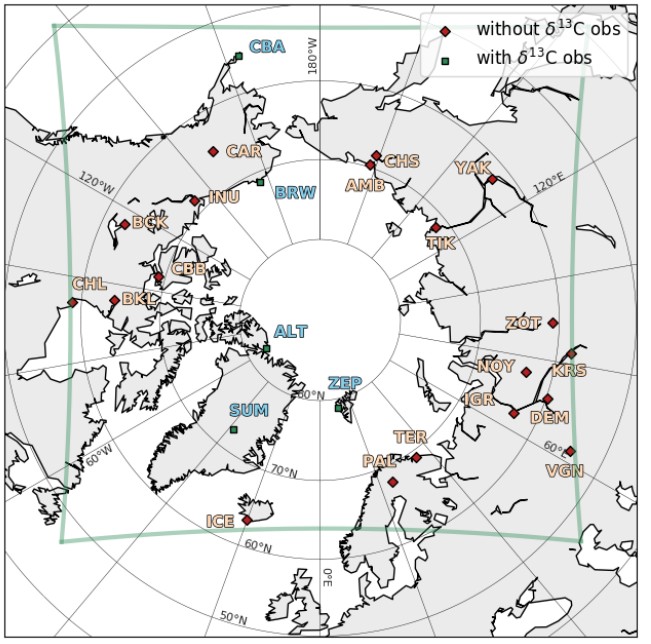


**Figure 1.** Delimitation of the studied polar domain (green line) and location of the 24 measurement sites used in this study and measuring atmospheric methane. Five stations (blue square) include flask measurements of $\delta^{13}$C-CH$_4$. The station name acronyms are given in Table 2.


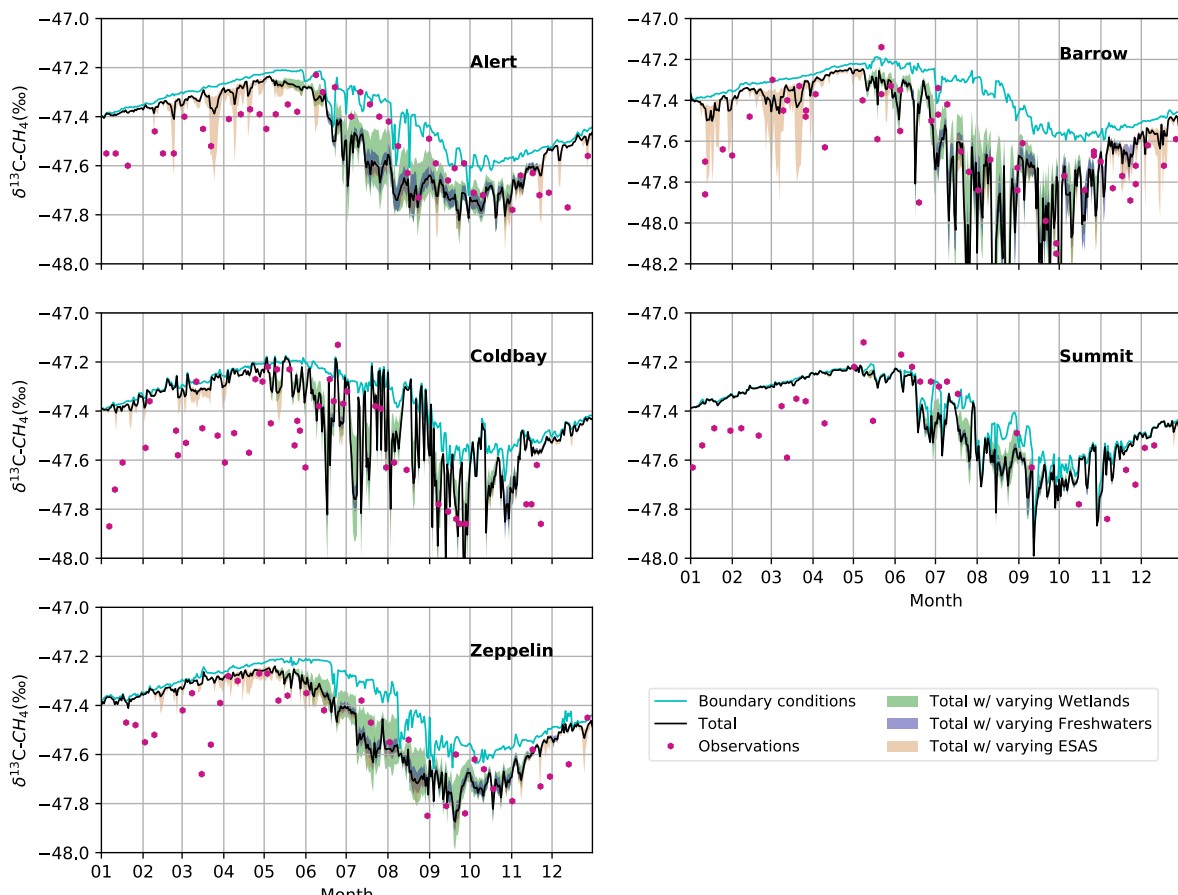


**Figure 2.** Time series of simulated and observed δ¹³C-CH₄, at five sites, in 2012. The cyan line represents the contribution of the boundary conditions; the black line represents the total simulated δ¹³C-CH₄ (boundary conditions + direct contribution of the sources located in the domain); the coloured shades represent total simulated δ¹³C-CH₄ with varying isotopic signatures (see Table 2) for wetlands (green), freshwater systems 1100 (blue) and ESAS (orange). The pink dots represent the flask observations. The hourly-simulated values are averaged into daily values. (Note the different vertical scale for Barrow: the minimum for simulations at Barrow exceeds the chosen scale and reaches -49.3‰.)

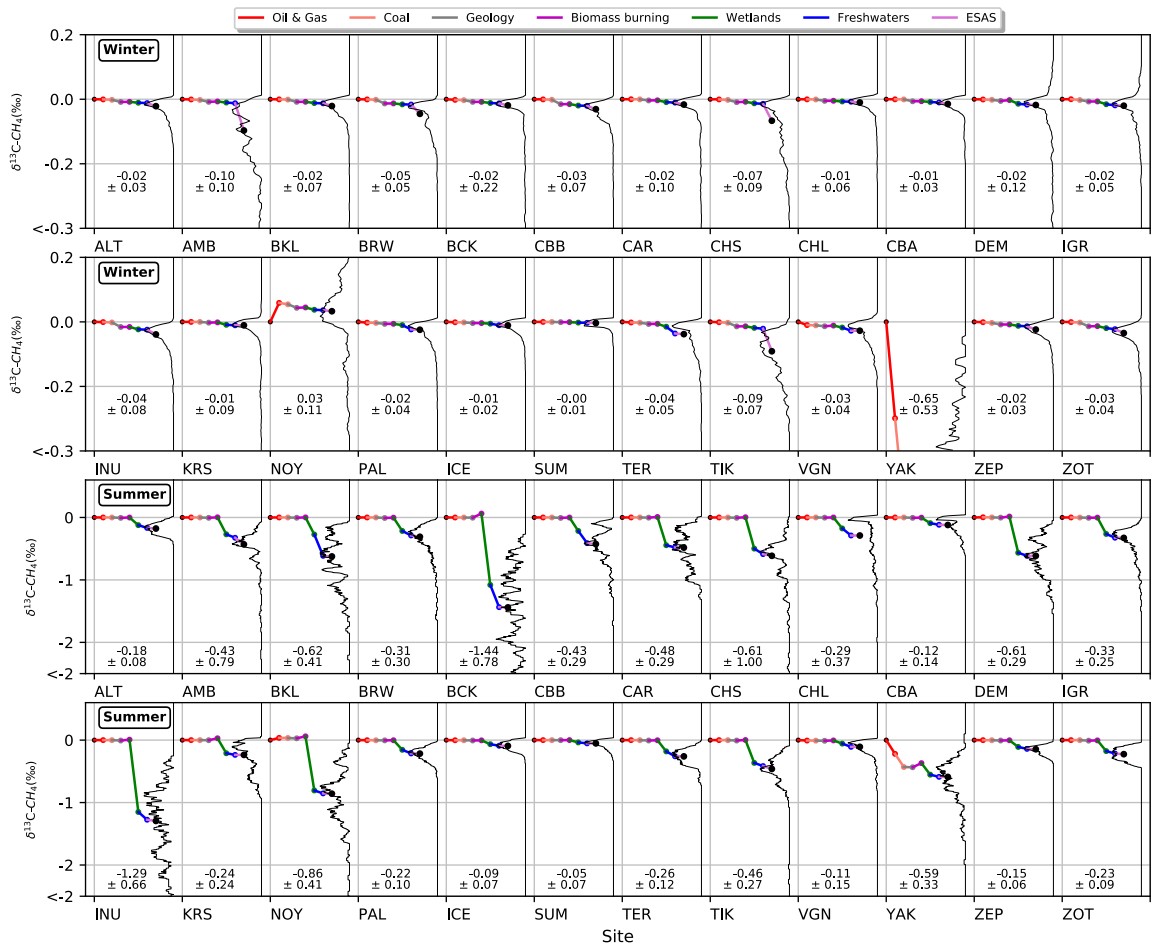


**Figure 3.** Winter (top two panels) and summer (bottom two panels) means of the direct contributions of the various northern high latitude sources to the $\delta^{13}$C-CH$_4$ value (in ‰) simulated by CHIMERE at 24 sites in 2012. The frequency distribution of daily signatures at each site is over plotted with an arbitrary unit on the x-axis, showing the simulated spread of the signal over the season. For each station and season, the number indicates the
mean $\delta^{13}$C-CH$_4$ value (in ‰) associated with its one-sigma value. See further details in Sect. 3.2.

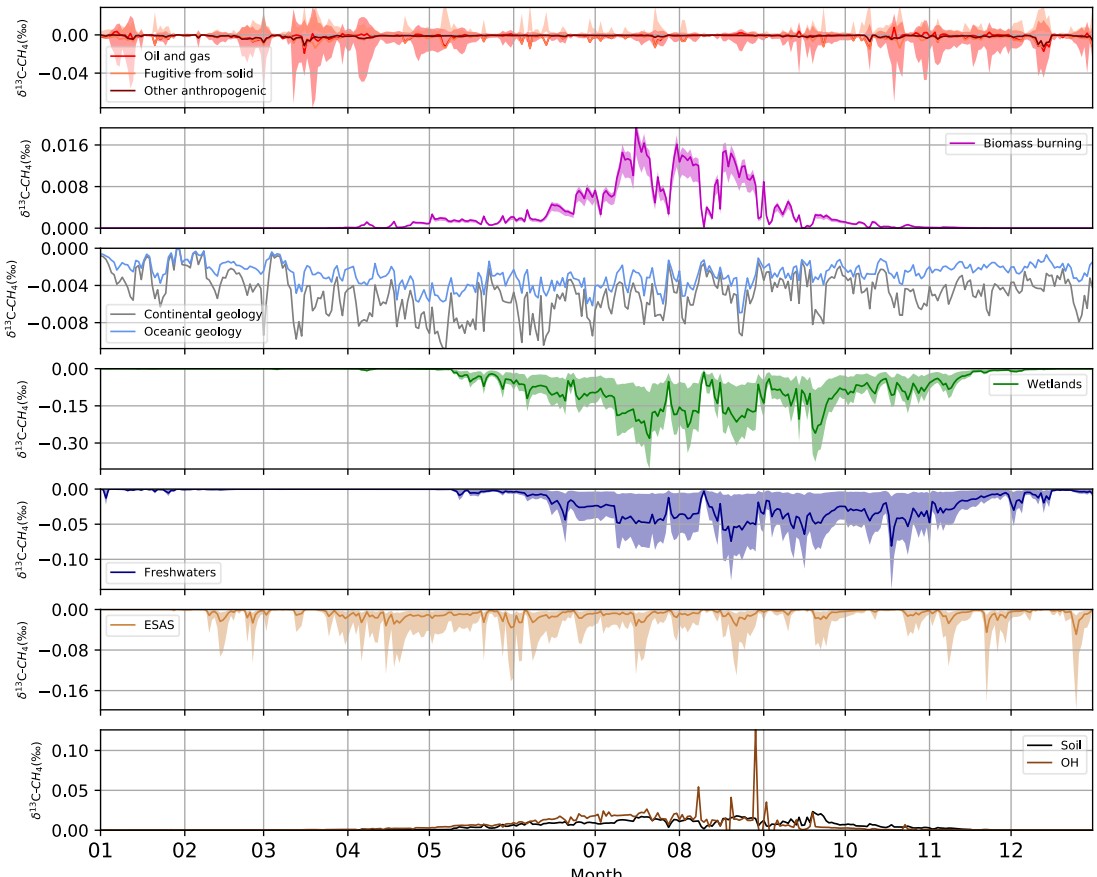

**Figure 4.** Time series of $\delta^{13}C$-$CH_4$ contribution of each source (in ‰), simulated by CHIMERE, in Zeppelin in 2012. The coloured shades represent the range of $\delta^{13}C$-$CH_4$ values when varying isotopic signatures (see Table 2). (Note the different scales.)

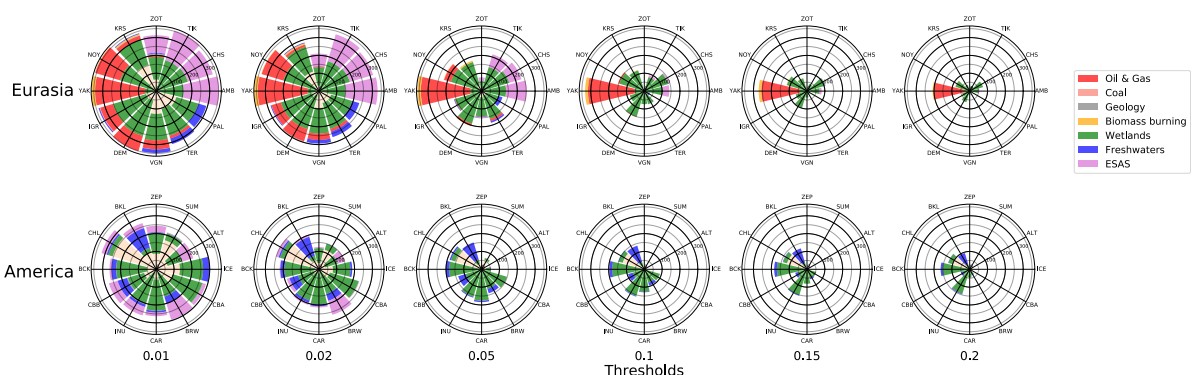

**Figure 5.** Number of days in 2012 when simulated daily direct contributions of northern high latitude sources to the $\delta^{13}C$-$CH_4$ value are above given thresholds, for each of the 24 stations of Fig.1 (Eurasia on top panel and America on bottom panel). The coloured shades indicate the dominant northern high latitude source in terms of $\delta^{13}C$-$CH_4$ contribution.