# Peer review of "Assessment of the theoretical limit in instrumental detectability of northern high-latitude methane sources using $\delta^{13}CH_4$ atmospheric signal"

_Atmospheric Chemistry and Physics, 2018_

## Referee Comment (RC1) · Anonymous Referee #1 · 29 Jan 2019

In this study, an atmospheric transport model is used to quantify the accuracy that is required for continuous measurements of d13C-CH4 in the Arctic. Isotopic measurements provide important process specific information about sources and sinks, which has proven very useful in global studies of methane using flask measurements from the global monitoring network. In recent years, instruments are becoming available for continuous measurements of methane isotopes. Their application is still limited, but has been demonstrated to be useful for regional networks. So far, however, they have yet not been deployed in the Arctic zone, but this could be a very promising application. This study quantifies the amplitude of the isotopic signals that can be expected, which is a useful contribution. I am less convinced about the approach to quantify detectabil-

ity, as will be discussed further below. Some suggestions are made to help improve that part, and widen the application area to further strengthen the significance of this work.

GENERAL COMMENTS

In my opinion, the scientific value of being able to detect methane emissions from wetlands in the Arctic is limited. We know that those emissions exist, and that they are important. More interesting is to be able to improve their quantification. For that, detection is not a sufficient requirement. The detection of regional trends would add significant understanding, but for that the requirements will be different. The question is not only about single measurement precision, but also the minimum number of measurement sites needed. This also brings in the dimension of data averaging, reducing the requirements depending in the statistics of the errors, the measurement frequency, and the temporal resolution that is needed. The conditions that are used to define 'detectability' in this study are not well motivated. Since the required measurement performance will depend on the details of the scientific questions that the measurement should help to answer, however, I think that to quantify the expected amplitude of variation is a more important outcome. It is possible to turn this into requirements, but then the purpose should be more clearly defined, and the inevitable limitations should be discussed as well.

An important distinction is found between remote, and regionally to locally influenced stations. Since the signal amplitudes differ between those sites, so will the measurement requirements. Yet the abstract and conclusion sections generalize the requirements to a single set. It should be made clearer what kind of sites are addressed by the numbers that are listed (rather than just a statement that the requirements will vary between sites). More useful would be to distinguish between applications. For some applications the requirements may be less stringent, especially if a larger number of cheaper sensors are deployed.

Over land, the amplitude of the signal will depend strongly on the altitude of the air inlet, and therefore the model level that is sampled. The altitudes in Table 1 probably refer more to the local orography than the height of the measurements with respect to the ground. There is a potential for increasing the significance of this work by adding the vertical dimension. What is the implication for required accuracy of towers and aircraft measurements?

SPECIFIC QUESTIONS

page 3, line 140: Although not long-term, the benefit of high frequency measurements was convincingly demonstrated by Roeckmann et al (acp, 2016).

page 4, line 218: It seems that the detectability of biomass burning could be influenced by the use of monthly average emissions, since in reality they may vary strongly with time.

page 4, line 224: GLOGOS

page 5, line 255: The d13C value of natural gas from West Siberia is known to be highly depleted (see e.g. Tarasova et al, 10.1007/s10874-010-9157-y)

page 7, line 366: 'However, they are excluded from our analysis . . .' But later the threshold detectability is defined from the source making the largest contribution to the signal. Shouldn't this signal include variations due to the background (it they overwhelm the regional sources this should limit the detectability)

page 8, line 441: Wouldn't the fact that the most significant sources all lead to methane depletion limit detectability. How do you distinguish one depleted source from another? It occurs to me that the definition of detectability ought to take differences in signatures into account, rather than only single process contributions.

Table 4: Is the year dependence of the thresholds important enough to restrict it to the year 2012?

Figure 3: What do the triplets of numbers at each site represent?

Figure 5: This shows that for a median wetland signature, the threshold of 0.5 per mil listed in the abstract would yield no single day of measurements. This seems to suggest that 0.5 is a too relaxed requirement.

TECHNICAL CORRECTIONS

Page 2, line 63: carbon dioxide

page 4, line 235: ERA-Interim reanalysis

Table 1: 'Range' i.o. 'Variant'
* * *

---

## Referee Comment (RC2) · Anonymous Referee #2 · 4 Feb 2019

GENERAL COMMENT Discriminating between methane sources is critical to determining regional and global emissions budgets and verifying national inventories. Mole fraction information is not enough – to determine sources, isotopic data are needed also. But isotopic measurement is difficult. It is analytically tough to attain both high precision and high sampling rate. Thonat et al. address this problem directly, by using modelling to ask how accurate the measurements need to be. The focus of the paper is the Arctic, but the same type of logic applies to all regions and to global measurement/modelling.

Thus the subject of the paper is important. The analysis that is presented makes a

valuable contribution towards design of a future measurement network. In detail, the paper is intelligently crafted and well presented. It is clearly laid out and the writing is straightforward and understandable. I have many comments on detail, but overall, the paper makes a valuable and worthwhile contribution to our knowledge and will be influential in planning future measurement systems.

To conclude, I recommend that the paper should be published, after some modifications.

SPECIFIC COMMENTS Title – missing 'the' before 13C. Might be better actually to say 'the $\delta$13CCH4Âň atmospheric signal"?

Line 13 First sentence of abstract is waffle. Delete. L 21 Specify that the study is about Carbon isotopes – D/H isn't mentioned. L 33 20% - could mention the more recent Etminan et al study that implies a larger number. Etminan, M., et al. (2016) Radiative forcing of carbon dioxide, methane and nitrous oxide: a significant revision of the methane radiative forcing. Geophys. Res. Lett. 43, 12,614–12,623, L41 Maybe mention Naus et al? Naus, Stijn, et al. (2019) Constraints and biases in a tropospheric two-box model of OH. Atmospheric Chemistry and Physics 19, 407-424. L 45 Nisbet et al. 2016 ?wrong year? L 46 and climate risk. L 50 – land thermokarst also? – e.g Yamal blowouts. There is also the wider problem of what is a natural wetland and what is a freshwater system. If the difference is in area of exposed water surface, then it's a bit like trying to determine who is the world's smallest giant. L 64 "compared to carbon dioxide's" – reads more easily if you delete the 's. Also maybe cite Kirschke et al here – I know it's mentioned later and you also cite Saunois, but seems appropriate here? L 81 – This is important – only 13C is considered. But either here or in the conclusion there should be a discussion of the potential value of restoring D/H measurement, and perhaps also a brief mention of clumped isotopes. L 85 notation - not possible to show in the constraints of acp online but a better notation might be d13CsubscriptCH4 L 107 – mention scarcity of D/H measurement. L113 – maybe cite Zazzeri et al here? The coal number is a real problem as Zazzeri found – increasingly open cast mining seems

to be emitting recently made biological methane coming from present day microbial activity on mine benches and this methane can be very light in C isotopes. L136 – 'permanently' increasing??? I used to think this 10 years ago, that optical instruments would soon catch up with mass spectrometry. But not so – if you want high precision (0.05‰ the optical methods need so much sample that the wind has changed by the time you complete the measurement on line, so you have to take grab samples, and then basically the cost and effort is comparable to mass specs. L150 paragraph – good plan! L160-170 Note that methane d13C is also measured in very long time series by NIWA-New Zealand, by the Japanese (e.g. Ny Alesund), and in Europe by RHUL, MPI and Utrecht. From memory, most labs have precision is rather better than 0.1 See Umezawa, T. et al. (2018) Intercomparisons of $\delta$13C and $\delta$D measurements of atmospheric CH4 for combined use of datasets from different laboratories. Atmos. Meas. Tech.., https://doi.org/10.5194/amt-2017-281 L186 – maybe say a little more about initial conditions? – Important. L197 – wetland/freshwater difference and soil negative source, etc etc. Needs a bit more detail. Maybe also mention Fisher et al (2017) L205 – CH4 emissions are limited in winter in the Arctic ???????. . .do you just mean wetland emissions? The way this is written implies that Russian gasfield emissions are trivial and can be written off as not important even before you do the study. Yet in the next paragraph you say anthropogenic emissions are >20Tg/yr, and we know much of the gasfield emission is in winter when the gas is being pumped most. L213 – EDGAR – here comes the top-down vs bottom-up problem. Needs to be discussed – you need to justify whet EDGAR is the least-worst option. L225 – note Petrenko et al, which strongly challenges the Etiope et al estimates. Petrenko, V.V. et al. (2017) Minimal geological methane emissions during the Younger Dryas–Preboreal abrupt warming event. Nature doi:10.1038/nature23316 L231 – 'prescribed' – this needs to be justified. Seems rather large. Again, what is a lake? What's the smallest giant? Why isn't a 1m2 puddle a lake? L250 Levin et al – -50 ‰ Russian gas. Note also Meth-MonitEUr report in which the St Petersburg team actually measured from a tower in a gasfield. EU Meth-MonitEUr Report Section 6 is online. -46‰ seems a bit heavy for Russia as I

have the sense that the production gas is isotopically lighter in the north. L276 Cattle – depends a lot on C4 (Maize, Sugar cane tops) or C3 (temperate hay, other feeds) diet. In the north, the likelihood is that much of the diet is C3 – the C4 grasses are mostly tropical or subtropical. C3 fed ruminants are probably more –ve in CH4. L285 -49‰ for geological – I'd query that. Most Arctic geological emission is hydrate and that is simply a storage vehicle for whatever rises into it. More like -50 to 55 per mil. But widely variable. Also see Petrenko et al cited above. L290 – -24 might be too heavy. Biomass burning in the boreal realm is entirely C3 plants and thus much lighter than tropical C4 grass fires. I'd take Chanton's values for northern US. L295 – wetlands – Arctic wetland methane source is entirely C3 and thus lighter than tropical C4 swamps – also methanotrophy. Agree with choice of Fisher and France et al values because aircraft sample an integrated signal over a wide area. But they did see a range of values. L322 – freshwater ambiguity again. L342 soil uptake 'equal to biomass burning' – no justification given. Can this be discussed? And bulk mass equality doesn't equal isotopic mass equality. L354 – no mention of the Cl sink. – Use Hossaini numbers? Hossaini, R., et al. (2016) A global model of tropospheric chlorine chemistry: Organic versus inorganic sources and impact on methane oxidation. Journal of Geophysical Research: Atmospheres 121.23 (2016). L360 Table 2 and L376 – note that Cold Bay is not Arctic. Average January Max T is near 1 degree C – above freezing. It's in the warm currents of the N Pacific. 55N – about the same as the chilly icebergs of the island of Sylt, Germany where folk paddle in swimsuits, and south of the deep frozen wastelands of Copenhagen and southernmost Sweden. L374 – the crosses for the data points. The use of crosses implies errors – but these don't look like the errors. The Time error is essentially zero. The measurement error is perhaps 0.06 per mil plus/minus. The data should be shown as vertical lines plus minus from the dot. L381. Boundary input – for Barrow I suspect the 2007 swing was from air that blew up from the boreal wetlands in mid-summer. L387 a 'depleted peak' is an oxymoron. Sounds like someone took a shovel to the top of Mt Everest and scooped off a few hundred metres. Better say 'spikes' throughout. Are the peaks 'observed' – i.e. real measurements? which data

show that: what are you classifying as a peak? Am I correct that you are saying that the various drops in the Barrow and Alert records are clearly caused by ESAS? Are you sure they are not just blips in a statistically thin data set? L390 – seasonality capture. Interesting, as Warwick had similar problems with capturing seasonality in her modelling. L400 – maybe a comment on the potential value of D/H also? L433 - -46‰ assumption – is that valid for the Arctic gasfields? What happens if you take a -50 per mil number as supported by Levin et al? The Korotchaevo tower measurements (increment ∼100 ppb) gave around -50 per mil during Sept. 2004 (Meth-MonitEUr report Section 6 – Reshetnikov team's results from a gasfield/wetland mix are -49.84 -52.43 -67.16 -65.14 -67.13 -53.49 -55.77 -49.30 depending on proportions of gas and wetland source. Accessible on web). L448 – maybe say 'more negative than' rather than 'less than' L462 – Zeppelin. Is this correct? – See France et al and Fisher et al. Note also that Zeppelin now has 5 samples a week analysed for d13C (MOCA project_NILU) L469 – varying the isotopic signatures... L486 – CL sink is small but has a large isotopic leverage – is this statement valid? Maybe cite Hossaini et at paper (see above). L493 – maybe cite Fisher et al 2006 – 0.05 per mil. Fisher, R., et al. (2006) High precision, automated stable isotopic analysis of atmospheric methane and carbon dioxide using continuous-flow isotope ratio mass spectrometry. Rapid communications in Mass Spectrometry, 20, 200-208. Note that the NIWA lines attain 0.03 per mil but with bigger samples. L499 – I'm rather sceptical of optical claims for 0.1 per mil precision in routine operation in remote settings. The cal gas demands would be extreme as the drift is hard to contain. 'Measurements are independent over the day' – but that means you integrate out your signal! Yes, if we mix all the paint in the world in one bucket we will get a very steady high-precision grey, but I rather like looking at colours in paintings. L517 – at ZEP the daily flask measurements are currently to 0.05 per mil. But there have been some contamination problems. L555 – spelling. Schaefer. L569 – basically this is saying that at the moment the high precision of mass spectrometry is needed to get a decent signal? L576-580 - Any thoughts on the usefulness of D/H?

Table 1- Cold Bay and Churchill are not Arctic, though I accept Churchill is pretty cool in winter. Cold Bay is maritime. Table 2 should give sources perhaps as a ref to Thonat 2017? Table 3 – note Fisher et al have Canadian results (-67±1 per mil) They have –66.8 ± 1.6‰ at East Trout Lake in Saskatchewan (Figure S4) and -67.2 ± 1.1 at Fraserdale, and Kuhlmann et al. 1998 had similar findings in Canada. Kuhlmann, A. J., Worthy, D. E. J., Trivett, N. B. A., & Levin, I. (1998). Methane emissions from a wetland region within the Hudson Bay Lowland: An atmospheric approach. Journal of Geophysical Research: Atmospheres, 103(D13), 16009-16016. Table 4 – is this the lowest detectability threshold? Or the highest? – i.e. the system has to be below this to spot the signal? 0.01 per mil for Teriberka ? I'm surprised – intuitively seems rather low? Fig 2 + for observations implies error bars – should be replaced by dots with error lines up and down. . .Time error is minimal. Fig 3 – a bit hard to see colours. Make sure the publication is large for this figure.

---

## Author Comment (AC1) · 12 Jul 2019

**"Assessment of the theoretical limit in instrumental detectability of Arctic methane sources using** 13**C atmospheric signal"** ***by*** **Thibaud Thonat et al.**

*Reviewers' comments are in italic blue.*
Responses are in normal black font. **Changes in the text are in black bold.**

**Response to Anonymous Referee #1** –

We are very grateful to Referee #1 to have reviewed the manuscript and submitted helpful comments and suggestions to improve both the study and the text.
Here we respond to the reviewer point by point.

*GENERAL COMMENTS*
*In my opinion, the scientific value of being able to detect methane emissions from wetlands in the Arctic is limited. We know that those emissions exist, and that they are important. More interesting is to be able to improve their quantification. For that, detection is not a sufficient requirement. The detection of regional trends would add significant understanding, but for that the requirements will be different. The question is not only about single measurement precision, but also the minimum number of measurement sites needed. This also brings in the dimension of data averaging, reducing the requirements depending in the statistics of the errors, the measurement frequency, and the temporal resolution that is needed. The conditions that are used to define 'detectability' in this study are not well motivated. Since the required measurement performance will depend on the details of the scientific questions that the measurement should help to answer, however, I think that to quantify the expected amplitude of variation is a more important outcome. It is possible to turn this into requirements, but then the purpose should be more clearly defined, and the inevitable limitations should be discussed as well.*
To address this first part of the general comment, we acknowledge that the detectability definition used in the submitted text (based on the signal departure from the background) was not suitable for observations analysis. We have redefined the "detectability" from an inverse modeling point of view. As a result we now analyze the amplitude of the variations of the total simulated signal – corresponding to what would be measured in the atmosphere. We compare the amplitude of the simulated signal to some instrument precision (called threshold). Then we determine which source (including boundary conditions) contributes the most to the variation in the simulated/expected signal. We acknowledge that this is a first order contribution as sources may overlap in time and space.
*An important distinction is found between remote, and regionally to locally influenced stations. Since the signal amplitudes differ between those sites, so will the measurement requirements. Yet the abstract and conclusion sections generalize the requirements to a single set. It should be made clearer what kinds of sites are addressed by the numbers that are listed (rather than just a statement that the requirements will vary between sites).*
This second part is also addressed in the revised manuscript:  we include variations in the signal due to boundary conditions and regional/local sources. Also Figure 5 has been modified and presents the potential detection at all

stations, allowing to have a quick look at which station is able to detect which source depending on the instrument uncertainty. We now include more discussion in the text and more details of the results in the abstract and conclusions.

*More useful would be to distinguish between applications. For some applications the requirements may be less stringent, especially if a larger number of cheaper sensors are deployed.*

This part has also been addressed when clarifying the detectability definition. We now clearly state that this study focus on signal that could help regional inverse modeling in better quantifying methane emissions. So that we define the detectability based on daily signal – used in regional inverse model.

*Over land, the amplitude of the signal will depend strongly on the altitude of the air inlet, and therefore the model level that is sampled. The altitudes in Table 1 probably refer more to the local orography than the height of the measurements with respect to the ground. There is a potential for increasing the significance of this work by adding the vertical dimension. What is the implication for required accuracy of towers and aircraft measurements?*

Yes the altitudes in Table 1 refer to the altitude of the station not of the air inlet. Here we use the inlet altitude corresponding to each existing site, associated with the corresponding vertical level of the model (as will be done for an atmospheric regional inversion), so this should include the existing tall towers. Using aircraft measurements is not really appropriate in our framework where we consider daily means.

SPECIFIC QUESTIONS

*page 3, line 140: Although not long-term, the benefit of high frequency measurements was convincingly demonstrated by Roeckmann et al (acp, 2016).*

This reference has been added to the text. "**For example, Röckmann et al. (2016) have deployed high frequency isotopic measurements of both $\delta^{13}$C-CH$_4$ and $\delta$D-CH$_4$ at Cabauw in Europe and were able to identify specific events and allocated them to specific anthropogenic sources (ruminants, natural gas or landfills)**."

*page 4, line 218: It seems that the detectability of biomass burning could be influenced by the use of monthly average emissions, since in reality they may vary strongly with time.*

Actually there was a typo in the text as we do use daily emissions from GFED and not monthly. So the detectability calculated here does take into account the strong temporal variation of biomass burning. "**monthly**" as been changed to "**daily**". However the signal of this source would also highly depend on the studied year as biomass burning has strong inter annual and spatial variability: we added a comment on this in Sect. 4: "**This study has been carried out only for the year 2012 as a test case. However, not all emissions have a high inter annual variability, such as does biomass burning. As a result, our findings should be still valid for the other sources for most of the years over a few future decades.**"

*page 4, line 224: GLOGOS*
Typo corrected

*page 5, line 255: The d13C value of natural gas from West Siberia is known to be highly depleted (see e.g. Tarasova et al, 10.1007/s10874-010-9157-y)*

We know include sensitivity tests to d13C signature for natural gas, and the isotopic signatures range between -40‰ to 50‰, with a mean value of -46‰. (**see text and Table 3).**

*page 7, line 366: 'However, they are excluded from our analysis …' But later the threshold detectability is defined from the source making the largest contribution to the signal. Shouldn't this signal include variations due to the background (it they overwhelm the regional sources this should limit the detectability).*

Indeed, the signal does include variations from the background, that is our boundary conditions here (lateral and top of the model). To address this and refine our analysis, we have first deleted this sentence and then changed the way we calculate detectability. "**Here we focus on a detectability definition taken from a regional inversion point of view: regional inversion systems analyse daily signals and optimize sources depending on synoptic deviations of the observed signals compared to the simulated ones. Therefore, a measuring instrument is considered to provide useful information to the inversion only if the synoptic variability of the atmospheric signal can be detected. To that end, we compute detectability capability in Fig. 5 and Tab. 4 as follows: (1) we compute the standard deviation over a five-day running window of the simulated total isotopic signal; (2) for a set of instrument precision threshold (from 0.2 to 0.01‰ see Fig. 5 and Tab. 4), if the running standard deviation is higher than the corresponding threshold, the source with the higher running standard deviation for the same 5-day window is considered detected for that one day; (3) for each threshold, we count the number of days over the year that each source is detected. Although the total atmospheric signal integrates contributions from different sources with different isotopic signatures, we keep only the major source contributing to the signal as a first order signal.**" In this way we are able to distinguish when the variation in the signal is due to the background (boundary conditions) or to regional sources**.** For some stations (such as Churchill), close to the border of the domain, the background contributes the most to the signal variations (new Fig. 5). This is further discussed in the revised manuscript.

*page 8, line 441: Wouldn't the fact that the most significant sources all lead to methane depletion limit detectability. How do you distinguish one depleted source from another? It occurs to me that the definition of detectability ought to take differences in signatures into account, rather than only single process contributions.*

Indeed the atmospheric signal integrates the contributions from the different sources. Here we select the source that contributes the most to the depletion though we acknowledge that several sources may simultaneously contribute. However discussing the overlapping in time and space of the sources is challenging without any real measurements as both the emission source and magnitude and the isotopic signatures are uncertain in the model. As a result, we present here a first order signal. After the definition of our detectability, we have included the following sentence:" **Although the total atmospheric signal**

**integrates contributions from different sources with different isotopic signatures, we keep only the major source contributing to the signal as a first order signal**."

We acknowledge that multiyear simulations may strengthen the results, especially if the year 2012 were specific for any reason. However this study is a test case and more efforts will be made as soon as continuous measurements are available (which should happen soon). We expect the year dependency being important mainly for biomass burning emission detection. In the discussion, we have added the following sentence:" **This study has been carried out only for the year 2012 as a test case. However, not all emissions have a high inter annual variability, as does biomass burning. As a result, our findings should be valid for the other sources for most of the years over a few future decades**."

*Figure 3: What do the triplets of numbers at each site represent?*
Figures 3 has been re-arranged to facilitate its reading. The triplets have disappeared. They indicated average, low and high range of total contributions to isotopic ratios.

*Figure 5: This shows that for a median wetland signature, the threshold of 0.5 per mil listed in the abstract would yield no single day of measurements. This seems to suggest that 0.5 is a too relaxed requirement.*
The conclusions in the abstract have been modified accordingly to the new definition of detectability. Also we detail more the results for the different types of stations.

*TECHNICAL CORRECTIONS*
*Page 2, line 63: carbon dioxide*
*page 4, line 235: ERA-Interim reanalysis*
*Table 2: 'Range' i.o. 'Variant'*
The technical corrections have been applied

---

## Author Comment (AC2) · 12 Jul 2019

**"Assessment of the theoretical limit in instrumental detectability of Arctic methane sources using** 13**C atmospheric signal"** *by* **Thibaud Thonat et al.**

*Reviewers' comments are in italic blue.*
Responses are in normal black font. **Changes in the text are in black bold.**

**Response to Anonymous Referee #2**

We are very grateful to Referee #2 to have reviewed the manuscript and submitted helpful comments and suggestions to improve the text.
Here we respond to the reviewer point by point.

*SPECIFIC COMMENTS*
*Title – missing 'the' before 13C. Might be better actually to say 'the δ13CCH4Ânˇ atmospheric signal"?*
Yes, the title has been modified accordingly

*Line 13 First sentence of abstract is waffle. Delete.*
This has been done

*L 21 Specify that the study is about Carbon isotopes – D/H isn't mentioned.*
This has been changed to « from methane isotopic $^{13}CH_4$ measurements".

*L 33 20% - could mention the more recent Etminan et al study that implies a larger number. Etminan, M., et al. (2016) Radiative forcing of carbon dioxide, methane and nitrous oxide: a significant revision of the methane radiative forcing. Geophys. Res. Lett. 43, 12,614–12,623,*
The reference has been added.

*L41 Maybe mention Naus et al? Naus, Stijn, et al. (2019) Constraints and biases in a tropospheric two-box model of OH. Atmospheric Chemistry and Physics 19, 407-424.*
This recently published manuscript has been added to refer to OH trends.

*L 45 Nisbet et al. 2016 ?wrong year?*
Indeed, the publication year is 2016.

*L 46 and climate risk.*
This suggestion has been included in the text.

*L 50 – land thermokarst also? – e.g Yamal blowouts. There is also the wider problem of what is a natural wetland and what is a freshwater system. If the difference is in area of exposed water surface, then it's a bit like trying to determine who is the world's smallest giant.*
We have added the land thermokarst sources as another source of interest in the region, associated with references to Wik et al. (2016).

*L 64 "compared to carbon dioxide's" – reads more easily if you delete the 's. Also maybe cite Kirschke et al here – I know it's mentioned later and you also cite Saunois, but seems appropriate here?*

The writing suggestion has been taken into account and we cite Saunois et al. (2016) for this sentence.

*L 81 – This is important – only 13C is considered. But either here or in the conclusion there should be a discussion of the potential value of restoring D/H measurement, and perhaps also a brief mention of clumped isotopes.*

We thank the reviewer for this comment. We have added the following comment « **Though measurements of $^{12}CH_3D$ exist, only $^{12}CH_4$ and $^{13}CH_4$ are considered in this study because they are the most abundant methane isotopologues in the atmosphere and as such are easier to measure than $^{12}CH_3D$. Regular measurements using flask samples exist since the early 2000s for $^{13}CH_4$. Unfortunately $^{12}CH_3D$, flask measurement series are scarce, with no published Arctic series for recent years. Laser spectrometer-based instrument for $^{13}CH_4$ continuous measurements are currently being or have been settled at different locations (e.g., Zeppelin mountain, Svalbarg, since 2018), while it is less the case for $^{12}CH_3D$ likely because only one instrument is commercially available.**"

*L 85 notation - not possible to show in the constraints of acp online but a better notation might be d13CsubscriptCH4*

We will see what is possible to do for the revised version or during the proof reading process.

*L 107 – mention scarcity of D/H measurement.*

We have mentioned the scarcity of D/H measurements earlier. This sentence is general and still true for 13CH4 measurements.

*L113 – maybe cite Zazzeri et al here? The coal number is a real problem as Zazzeri found – increasingly open cast mining seems to be emitting recently made biological methane coming from present day microbial activity on mine benches and this methane can be very light in C isotopes.*

We added the following sentence: "**Regarding coal emissions, Zazzeri et al. (2016) pointed out that global model usually use a signature of -35‰ for coal, while measurements show values between -30‰ and -60 ‰ depending on the coal type and depth (from anthracite to bituminous)**."

*L136 – 'permanently' increasing??? I used to think this 10 years ago, that optical instruments would soon catch up with mass spectrometry. But not so – if you want high precision (0.05‰ the optical methods need so much sample that the wind has changed by the time you complete the measurement on line, so you have to take grab samples, and then basically the cost and effort is comparable to mass specs.)*

Indeed… we have change this to "**satisfying performances**"

*L150 paragraph – good plan!*

*L160-170 Note that methane d13C is also measured in very long time series by NIWA-New Zealand, by the Japanese (e.g. Ny Alesund), and in Europe by RHUL, MPI*

*and Utrecht. From memory, most labs have precision is rather better than 0.1 See Umezawa, T. et al. (2018) Intercomparisons of δ13C and δD measurements of atmospheric CH4 for combined use of datasets from different laboratories. Atmos. Meas. Tech.., https://doi.org/10.5194/amt-2017-281*

Looking through Umezawa et al., the precision reached by the different laboratories range between 0.05 and 0.1 per mil for d13C. INSTARR precision is 0.08 per mil. We thank the referee for his comment on other available data set outside our domain. Regarding the NIWA data in Ny Alesund, including data from another laboratory would add calibrating issues between the networks, as Umezawa et al. show that laboratory spread ranges at 0.5 per mil for d13C.

*L186 – maybe say a little more about initial conditions? – Important.*

We added a sentence explaining a bit the set-up of this global simulation:

**"This global simulation used on ensemble of emission fluxes (including ORCHIDEE for wetland and EDGARv4.2 for anthropogenic and GFED4.1 for biomass burning emissions) that were adjusted in order to obtain a reasonable agreement at the global scale between the simulated isotopic signal and the flask measurements over the NOAA network."**

*L197 – wetland/freshwater difference and soil negative source, etc etc. Needs a bit more detail. Maybe also mention Fisher et al (2017)*

This paragraph aims at describing the modeling methodology. Definition and references describing each category is given in Section 2.3. We have added the following sentence:" **More details on the aforementioned emission categories are given below in Section 2.3**., as well as "**soil uptake, considered as a negative source at the surface**"

*L205 – CH4 emissions are limited in winter in the Arctic ???????. . .do you just mean wetland emissions? The way this is written implies that Russian gas field emissions are trivial and can be written off as not important even before you do the study. Yet in the next paragraph you say anthropogenic emissions are >20Tg/yr, and we know much of the gas field emission is in winter when the gas is being pumped most.*

Indeed, this was poorly written. We have reformulated to: "**No pair of tracers is implemented for the initial conditions: simulations in January are partly influenced by prescribed initial conditions from global fields during the spin up period of 2-4 weeks (typical mixing time of air masses in the domain with the chosen model set-up spanning high northern latitude regions) but this has little impact on our conclusions.**"

*L213 – EDGAR – here comes the top-down vs bottom-up problem. Needs to be discussed – you need to justify whet EDGAR is the least-worst option.*

Here we used the EDGAR inventory for consistency with the global simulation used as initial and boundary conditions, as well as with the first part of the study (Thonat et al., 2017). There might be discrepancies between top-down and bottom-up estimates in the anthropogenic emissions in the northern latitude regions. We do not perform any inversion of the signal, but forward simulation to assess the expected amplitude in the isotopic signal and whether this can be captured by the instruments and if so, which source could be distinguished.

Other inventories could have been tested (ECLISPE from GAINS, newest EGDRAv432 – not available when this study started), however anthropogenic emissions would be detected at the same sites as those found here (Russian cites closer to anthropogenic activities), with, probably, same detection thresholds.

*L225 – note Petrenko et al, which strongly challenges the Etiope et al estimates. Petrenko, V.V. et al. (2017) Minimal geological methane emissions during the Younger Dryas–Preboreal abrupt warming event. Nature doi:10.1038/nature23316*

Petrenko et al. (2017) suggests much lower geological estimates than Etiope et al., from 0 to less than 18 Tg/yr globally. Zero is probably non realistic given methane emissions from geological sources have actually been observed. The 18 Tg/yr is challenging not only to Etiope's bottom up estimates but also to top-down estimates. Further assessments of the geological emissions are needed for the methane budget (globally and regionally), but stand beyond this work. In this study, for consistency with Thonat et al. (2017), we keep the same inventory and emission estimates than in the first part of the study.

*L231 – 'prescribed' – this needs to be justified. Seems rather large. Again, what is a lake? What's the smallest giant? Why isn't a 1m2 puddle a lake?*

Indeed, we acknowledge that definitions of the different freshwater systems and their frontiers remain a tricky issue, still highly debated in the community. However solving this issue is far beyond the scope of this atmospheric modeling study. Here we rely on a global data set, GLWD, with its limitations. Improvement and agreement within the community on the frontier between lakes/ponds/puddles and their respective areas and contributions in a grid pixel (and their methane density fluxes) will definitely be a big step forward for the atmospheric modeling community using such data sets as input to their model. Meanwhile, we have to do our best from available data sets.
We have rephrased the first sentence to:" **Following Thonat et al. (2017), we considered that 15 TgCH$_4$ yr$^{-1}$ are emitted from all lakes and reservoirs located at latitudes above 50°N.**"

*L250 Levin et al – -50 ‰ Russian gas. Note also Meth-MonitEUr report in which the St Petersburg team actually measured from a tower in a gasfield. EU Meth-MonitEUr Report Section 6 is online. -46‰ seems a bit heavy for Russia as I have the sense that the production gas is isotopically lighter in the north.*

To address this comment and a similar comment from Reviewer#1, we now include tests over a range of isotopic signature for gas emissions (between -40‰ and -50‰, see Table 3 and shaded areas in Fig 4).

*L276 Cattle – depends a lot on C4 (Maize, Sugar cane tops) or C3 (temperate hay, other feeds) diet. In the north, the likelihood is that much of the diet is C3 – the C4 grasses are mostly tropical or subtropical. C3 fed ruminants are probably more –ve in CH4.*

Indeed, more C3 fed is expected for the high latitudes. A recent publication (to be published) suggests -67 per mil for Russia and -65 for North America. These values are lower than the one used here, -62 per mil. However, as these emissions do not contribute much to anthropogenic emissions (1.3 Tg against

more than 15 Tg for oil, gas and coal emissions), modifying the isotopic signature does not change the results (i.e. this category is not detected at the studied stations, see Figure 5). We have added the following sentence:"**The emissions of those two sources are an order of magnitude lower than anthropogenic emissions from fossil fuel production, changing their isotopic signature does not yield to higher isotopic signal than these of fossil fuel emissions.**"

*L285 -49‰ for geological – I'd query that. Most Arctic geological emission is hydrate and that is simply a storage vehicle for whatever rises into it. More like -50 to 55 per mil. But widely variable. Also see Petrenko et al cited above.*
For geological emissions, we have modified the isotopic signature and now use -52 per mil (as a medium value between -50 and -55 per mil). Updated text: "**In this region, geological manifestations occur through submarine seepages and microseepages with mean isotopic signatures of about -51.2‰ and -51.4‰ with uncertainty in the order of 7‰ and 2‰, respectively (Etiope et al., 2019). As a consequence, the isotopic signature used here for geological methane, both continental and submarine, is -52‰, following Etiope et al. (2019), associated to the range -50‰ to -55‰.**" The results show that the signal is about 0.001 ‰ (see Fig 3 and Supplementary), and is not detected with the considered isotopic signature (Fig 5).

*L290 – -24 might be too heavy. Biomass burning in the boreal realm is entirely C3 plants and thus much lighter than tropical C4 grass fires. I'd take Chanton's values for northern US.*
To address this comment, we now include tests over a range of isotopic signature for biomass burning emissions (between -21‰ and -30‰, see Table 3 and shaded areas in Fig 4).

*L295 – wetlands – Arctic wetland methane source is entirely C3 and thus lighter than tropical C4 swamps – also methanotrophy. Agree with choice of Fisher and France et al values because aircraft sample an integrated signal over a wide area. But they did see a range of values.*
Thank you for this comment.

*L322 – freshwater ambiguity again.*
We acknowledge that this word could be associated to many different water systems. We have added "**lakes and reservoirs**" in parenthesis after "freshwater system", as these are the systems taken into account here.

*L342 soil uptake 'equal to biomass burning' – no justification given. Can this be discussed? And bulk mass equality doesn't equal isotopic mass equality.*
Thank you for this comment. There is, indeed, no reason to compare the soil uptake with biomass burning emissions, even in magnitude (except to say that they cancel each other on a yearly basis). This has been rephrased to "**its magnitude is equal to -3.1 Tg CH$_4$ yr$^{-1}$ (see Table 2)**"

*L354 – no mention of the Cl sink. – Use Hossaini numbers? Hossaini, R., et al. (2016) A global model of tropospheric chlorine chemistry: Organic versus in- organic*

*sources and impact on methane oxidation. Journal of Geophysical Research: Atmospheres 121.23 (2016).*

Indeed, our simulation did not include any chlorine oxidation. We have shown in Thonat et al., 2017, that Cl sink in the regional simulation has a negligible impact on $CH_4$ mixing ratios (below 1ppb because of the relatively short time residence of air masses in our domain of simulation). Also there have been a number of studies finding that the tropospheric chlorine sink has been overestimated. Wang et al. (2017) suggests about 5Tg/yr globally instead if 12-13 Tg/yr in Hossaini. Gromov et al. (2018) lowered this value to 1Tg/yr.

Although the isotopic fractionation is larger through chlorine oxidation than through OH oxidation, due to higher KIE, we expect a rather small impact on 13CH4, considering the methane lifetime against Cl – in our regional simulation. Also any effect from this sink would need to be simulated in the global model serving as boundary conditions. This would add some very large-scale signal to the boundary conditions, probably limited though. Anyway, we think this will not change the results on the detectability of the regional Arctic sources. We have added the following text in the revised manuscript: "**The chlorine sink is not included in our regional simulation. We have shown in Thonat et al., 2017 that this sink has a negligible impact of $CH_4$ mixing ratio (below 1ppb). Despite a high KIE, including this sink in the regional simulation will not change significantly our conclusions on the local source detectability.**"

*L360 Table 2 and L376 – note that Cold Bay is not Arctic. Average January Max T is near 1 degree C – above freezing. It's in the warm currents of the N Pacific. 55N – about the same as the chilly icebergs of the island of Sylt, Germany where folk paddle in swimsuits, and south of the deep frozen wastelands of Copenhagen and southernmost Sweden.*

Indeed, our domain extends further south than the Arctic region. We have taken into account this fair comment and now mention "**Northern high-latitudes" instead of "Arctic".** Here, in the title and elsewhere in the text and table where necessary.

*L374 – the crosses for the data points. The use of crosses implies errors – but these don't look like the errors. The Time error is essentially zero. The measurement error is perhaps 0.06 per mil plus/minus. The data should be shown as vertical lines plus minus from the dot.*

Fig. 2 has been modified accordingly, and crosses have been replaced by dots.

*L381. Boundary input – for Barrow I suspect the 2007 swing was from air that blew up from the boreal wetlands in mid-summer.*

Indeed, Fig S4 shows large contribution from wetland and freshwater emissions over these 3 months (about -0.5 per mil and -0.2 per mil respectively). These contributions are much higher than those simulated at the four other sites (about 0.2 per mil and 0.05 per mil). We have added the following sentence:" **Barrow is more sensitive to the regional sources (mainly wetland and freshwater emissions) compared to the four other sites (see Fig S4 against Fig 4, S1, S10 and S18).**"

*L387 a 'depleted peak' is an oxymoron. Sounds like someone took a shovel to the top of Mt Everest and scooped off a few hundred metres. Better say 'spikes' throughout. Are the peaks 'observed' – i.e. real measurements? which data show that: what are you classifying as a peak? Am I correct that you are saying that the various drops in the Barrow and Alert records are clearly caused by ESAS? Are you sure they are not just blips in a statistically thin data set?*

"Peak" has been replaced by spikes throughout. Here we are referring to the simulated signal. Indeed, it is hard to believe in real spikes in such low frequency data set. After some deletion, the text has been modified as follows: "**Nevertheless, large spikes are simulated in winter at Barrow and Alert, some of which are attributed to ESAS emissions. Due to the low frequency of flask measurements, it is hard to associate these simulated spikes to observed ones. Higher frequency measurements are needed to assess the reality of such spikes and their magnitudes, and to allow discussion on both the magnitude of the source and its isotopic signature.**"

*L390 – seasonality capture. Interesting, as Warwick had similar problems with capturing seasonality in her modelling.*

Indeed, we have modified the text as follows:"**The decrease in early summer comes too soon and so does the autumn minimum, as already noticed by Warwick et al. (2016).**"

*L400 – maybe a comment on the potential value of D/H also?*

Here we have just added "**as well as in $\delta D\text{-}CH_4$**", though the study focuses only on $\delta^{13}C\text{-}CH_4$ signal.

*L433 - -46‰ assumption – is that valid for the Arctic gasfields? What happens if you take a -50 per mil number as supported by Levin et al? The Korotchaevo tower measurements (increment ≈ 100 ppb) gave around -50 per mil during Sept. 2004 (Meth-MonitEUr report Section 6 – Reshetnikov team's results from a gasfield/wetland mix are -49.84 -52.43 -67.16 -65.14 -67.13 -53.49 -55.77 -49.30 depending on proportions of gas and wetland source. Accessible on web).*

To address this comment and a similar comment from Reviewer#1, we now include tests over a range of isotopic signature for gas emissions (between -40‰ and -50‰, see Table 3 and shaded areas in Fig 4).

*L448 – maybe say 'more negative than' rather than 'less than'*

This has been corrected.

*L462 – Zeppelin. Is this correct? – See France et al and Fisher et al. Note also that Zeppelin now has 5 samples a week analysed for d13C (MOCA project_NILU)*

This has been rephrased to:" **Zeppelin is a typical example of a remote site**."
Such recent measurements would be interesting to compare with simulations covering the recent years, as well as with the continuous measurements taking place there for more than one year now.

*L469 – varying the isotopic signatures...*

This has been corrected

*L486 – CL sink is small but has a large isotopic leverage – is this statement valid? Maybe cite Hossaini et at paper (see above).*

The Cl sink has a negligible impact on CH4 (less than 1 ppb at the surface, Thonat et al., 2017). The impact of chlorine oxidation on CH4 has been debated recently, with studies stating that the sink is probably overestimated in Hossaini et al. (2017) (see previous answer ).

*L493 – maybe cite Fisher et al 2006 – 0.05 per mil. Fisher, R., et al. (2006) High precision, automated stable isotopic analysis of atmospheric methane and carbon dioxide using continuous-flow isotope ratio mass spectrometry. Rapid communications in Mass Spectrometry, 20, 200-208. Note that the NIWA lines attain 0.03 per mil but with bigger samples.*

This reference has been added in the text: "**Using continuous-flow isotope ratio mass spectrometry, Fisher et al. (2006) reached a precision of 0.05 ‰.**"

*L499 – I'm rather sceptical of optical claims for 0.1 per mil precision in routine operation in remote settings. The cal gas demands would be extreme as the drift is hard to contain.*

We fully agree with this comment. This is indeed the next sentence "**However, Aerodyne instruments face a strong drift that imposes a strict calibration protocol (every 2 hours in most recent set-ups), which dramatically reduces the daily number of available observations to typically a few tens**"

*'Measurements are independent over the day' – but that means you integrate out your signal! Yes, if we mix all the paint in the world in one bucket we will get a very steady high-precision grey, but I rather like looking at colours in paintings.*

We choose to integrate the isotopic signal at the daily scale because the scope of the article is to pave the way towards regional inversions using isotopic ratios. In such systems, only the daily signal can be used, due to the transport model resolution.
We agree that continuous isotopic measurements could detect sub-daily signal coming from local sources, which could be very valuable for the vegetation process community for instance.

*L517 – at ZEP the daily flask measurements are currently to 0.05 per mil. But there have been some contamination problems.*

Thank you for this note. We have modified the text as follows:" **Currently, daily flask are operated at ZEP with an uncertainty of 0.05‰ but contamination problems occur. If such contaminations are avoided so that the measurement uncertainty reaches 0.05‰, some wetland events may be detected during about 10 days.**"

*L555 – spelling. Schaefer.*
This has been corrected.

*L569 – basically this is saying that at the moment the high precision of mass spectrometry is needed to get a decent signal?*

Lower precisions might be sufficient to study very small scale spikes linked to local emissions nearby one site, but in our regional inversion framework, it is true that our conclusion points at precision requirements only fulfilled by mass spectrometry so far.

*L576-580 - Any thoughts on the usefulness of D/H?*
Delta-D-CH4, may be useful to study the sinks as oxidation is fractionating in D/H. However such assessment needs to be carefully taken into account at the global scale in the model feeding the boundaries of the regional model, which has not been done in our group. Furthermore less data (observations and isotopic signatures) are available to evaluate the models and their sensitivity to smaller signals (than for DeltaC13 -CH4). We have open the perspectives in the conclusions.

*Table 1- Cold Bay and Churchill are not Arctic, though I accept Churchill is pretty cool in winter. Cold Bay is maritime.*
The title has been modified to "northern high latitude" instead of "Arctic", as well as elsewhere necessary in the text, Table and Figures.

*Table 2 should give sources perhaps as a ref to Thonat 2017?*
A sentence has been added in the caption: "**Methane emissions and isotopic signatures in the studied domain (see text, Sect. 2.3 and 2.4). Emission and sink fluxes used here are the same as in Thonat et al. (2017)."**

*Table 3 – note Fisher et al have Canadian results (-67±1 per mil) They have –66.8 ± 1.6‰ at East Trout Lake in Saskatchewan (Figure S4) and -67.2 ± 1.1 at Fraserdale, and Kuhlmann et al. 1998 had similar findings in Canada. Kuhlmann, A. J., Worthy, D. E. J., Trivett, N. B. A., & Levin, I. (1998). Methane emissions from a wetland region within the Hudson Bay Lowland: An atmospheric approach. Journal of Geophysical Research: Atmospheres, 103(D13), 16009-16016.*
In Kuhlmann et al. (1998), they found an isotopic signature of -60 per mil for wetland, as stated in Table 4. This missing reference has been added. The two values from the supplementary of Fisher et al., 2017 have been added to Table 4.

*Table 4 – is this the lowest detectability threshold? Or the highest? – i.e. the system has to be below this to spot the signal? 0.01 per mil for Teriberka ? I'm surprised – intuitively seems rather low?*
We have changed "lowest detectability" to "**minimum detectability**".
For Teriberka, the new detection definition gives 0.02 as minimum uncertainty.

*Fig 2 + for observations implies error bars – should be replaced by dots with error lines up and down. . .Time error is minimal.*
Figure has been modified where crosses have been replaced by dots.

*Fig 3 – a bit hard to see colours. Make sure the publication is large for this figure.*
Figure 3 has been modified. We will pay attention to the quality during the proof reading process and with the editor.